# Tensor-Var: Efficient Four-Dimensional Variational Data Assimilation

**Yiming Yang** [1]  **Xiaoyuan Cheng** [1]  **Daniel Giles** [1]  **Sibo Cheng** [2]  **Yi He** [1]  **Xiao Xue** [1]  **Boli Chen** [1]  **Yukun Hu** [1]

## Abstract

Variational data assimilation estimates the dynamical system states by minimizing a cost function that fits the numerical models with the observational data. Although four-dimensional variational assimilation (4D-Var) is widely used, it faces high computational costs in complex nonlinear systems and depends on imperfect state-observation mappings. Deep learning (DL) offers more expressive approximators, while integrating DL models into 4D-Var is challenging due to their nonlinearities and lack of theoretical guarantees in assimilation results. In this paper, we propose *Tensor-Var*, a novel framework that integrates kernel conditional mean embedding (CME) with 4D-Var to linearize nonlinear dynamics, achieving convex optimization in a learned feature space. Moreover, our method provides a new perspective for solving 4D-Var in a linear way, offering theoretical guarantees of consistent assimilation results between the original and feature spaces. To handle large-scale problems, we propose a method to learn deep features (DFs) using neural networks within the Tensor-Var framework. Experiments on chaotic systems and global weather prediction with real-time observations show that Tensor-Var outperforms conventional and DL hybrid 4D-Var baselines in accuracy while achieving a 10- to 20-fold speed improvement.

## 1. Introduction

Forecasting of dynamical systems is an initial value problem of practical significance. Many real-world systems, such as the ocean and atmosphere, are *chaotic*, which means that minor errors in current estimations in computational models can lead to rapid divergence and substantial forecast errors

[1]University College London, London, United Kingdom [2]CEREA, ENPC and EDF R&D, Institut Polytechnique de Paris, Paris, France. Correspondence to: Yiming Yang <zcahyy1@ucl.ac.uk>, Yukun Hu <yukun.hu@ucl.ac.uk>.

*Proceedings of the $42^{nd}$ International Conference on Machine Learning*, Vancouver, Canada. PMLR 267, 2025. Copyright 2025 by the author(s).

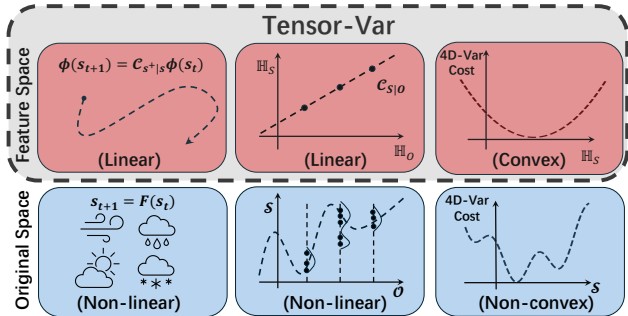

*Figure 1.* Demonstration of Tensor-Var: A DA system with nonlinear dynamical model, observation model, and non-convex cost function (bottom) can be represented linearly in feature space, resulting in a convex cost function (top).

([Coveney, 2024](#)). In this regard, data assimilation (DA) ([Law et al., 2015](#); [Asch et al., 2016](#)) uses observation data to continuously calibrate models, improving forecast accuracy. Various DA methods have been proposed to deal with different types of observation data and system dynamics. Among these methods, 4D variational (4D-Var) data assimilation has been considered cutting-edge and used effectively in real-world applications such as numerical weather prediction (NWP) ([Browne et al., 2019](#); [Milan et al., 2020](#)). 4D-Var minimizes a quadratic cost function that finds the optimal match between system states and observations ([Asch et al., 2016](#)). While effective in NWP, there are two critical limitations for their applications: (1) Numerical models for complex, nonlinear systems are often inefficient for real-time assimilation and forecasting. (2) Observations are often noisy, incomplete representations of the states, or even without a known state-observation mapping, posing challenges in utilizing the observations.

Efforts have been made to integrate DL models to learn an observation (or inverse) model ([Frerix et al., 2021](#); [Wang et al., 2022](#); [Liang et al., 2023](#)), addressing the imperfect knowledge of observation models in 4D-Var. While these approaches improve observation utilization, they remain constrained by the complexities of numerical models and learned observation mappings, whereas our approach simplifies them by finding their linear representations. To improve computation efficiency, state-of-the-art DL models ([Vaswani et al., 2017](#); [Chen et al., 2018](#); [Li et al., 2020](#); [Kovachki et al., 2023](#); [Cheng et al., 2025](#); [He et al., 2025](#)) are capable of con-

structing highly nonlinear mappings to surrogate dynamical systems and achieve notable successes in NWP (Bi et al., 2022; Lam et al., 2022; Kurth et al., 2023; Chen et al., 2023; Conti, 2024; Vaughan et al., 2024). However, integrating such models into optimization-based tasks, such as 4D-Var, remains challenging due to their inherent non-linearities (Janner et al., 2021; Bocquet, 2023; Bocquet et al., 2024). Using auto-differentiation (AD) of DL models in 4D-Var can reduce computational costs and has shown success in simple examples (Geer, 2021; Dong et al., 2022; Cheng et al., 2024). However, concerns remain regarding the accuracy of AD-derived derivatives, and their computational complexity increases with the model scale (Baydin et al., 2018). Recently, (Xiao et al., 2024) applied AD of a pre-trained weather forecasting model in 4D-Var, forming a self-contained DA framework for Global NWP. This approach demonstrates promising performance in large-scale settings, but relies on a well-designed pre-trained model and may not generalize easily to other domains. Latent DA (Peyron et al., 2021; Fablet et al., 2021; Melinc & Zaplotnik, 2023; Cheng et al., 2023a; Fablet et al., 2023) addresses these challenges by performing DA in a learned low-dimensional latent space. Although efficient, these approaches lack theoretical guarantees for the consistency of 4D-Var solutions between the latent and original spaces.

In this paper, we introduce Tensor-Var, a framework that linearizes nonlinear dynamics via kernel embedding, enabling convex optimization in feature space. Our approach addresses efficiency and non-convex challenges in existing variational DA methods. To our knowledge, Tensor-Var is the first attempt to identify the linear feature space for DA systems with a convex cost function, enabling an efficient solution to the 4D-Var problem. To handle incomplete observations, we derive an inverse observation operator that incorporates historical observations to infer the system states. Moreover, we provide a theoretical analysis that demonstrates the existence of a linear representation of the system under kernel features and the consistency of the 4D-Var solution across original and feature spaces. A key challenge in extending the kernel embedding to practical variational DA is scalability. To overcome this, our approach learns adaptive deep features (DFs) that map data into a fixed-dimensional feature space, reducing computational complexities. Our experiments on two chaotic systems and two global NWP applications show that Tensor-Var outperforms both conventional and ML-hybrid variational DA baselines in accuracy and computational efficiency, demonstrating the advantages of linearizing DA systems.

## 2. Preliminary

**Notation.** Let $S$ and $O$ be random variables representing the state and observation, respectively, with their realizations $s$

and $o$ taking values in the spaces $\mathcal{S} \subset \mathbb{R}^{n_s}$ and $\mathcal{O} \subset \mathbb{R}^{n_o}$, where $n_s$ and $n_o$ denote the dimensions of these spaces. The distribution of $S$ is denoted by $\mathbb{P}_S$, the joint distribution over $(S, O)$ by $\mathbb{P}_{SO}$, and the conditional distribution of $S$ given $O$ by $\mathbb{P}_{S|O}$. A sequence of states over time steps from 1 to $t$ is denoted by $s_{1:t} = (s_1, \ldots, s_t)$.

### 2.1. 4D variational data assimilation

Consider a dynamical system in discrete-time comprising a dynamical model and observation model:

$$s_t = F(s_{t-1}) + \epsilon_t^s, \text{ and } o_t = G(s_t) + \epsilon_t^o, \quad (1)$$

in which $F$ is the dynamical model that advances the state $s_t$ to $s_{t+1}$, and $G$ is the observation model that maps the $s_t$ to the observation $o_t$. The noise components $\epsilon_t^s, \epsilon_t^o$ such that $\epsilon_t^s \sim \mathcal{N}(0, Q)$ and $\epsilon_t^o \sim \mathcal{N}(0, R)$. The objective of 4D-variational DA is to minimize a cost function:

$$J(s_{0:T}) = \|s_0 - s_0^b\|_{B^{-1}}^2 + \sum_{t=0}^{T} \|o_t - G(s_t)\|_{R^{-1}}^2$$
$$+ \sum_{t=1}^{T} \|s_t - F(s_{t-1})\|_{Q^{-1}}^2, \quad (2)$$

in which $s_0^b$ is a prior guess for the initial state $s_0$, with $B$ as the background covariance matrix representing the uncertainty, i.e., $s_0 \sim \mathcal{N}(s_0^b, B)$. The second and third terms account for errors in the observation and dynamical models in (1). Solving the minimization (2) in a high-dimensional nonlinear dynamical system with incomplete observations can be challenging. In this paper, we aim to characterize the nonlinear dynamical and observation models in (1) using linear models, allowing efficient optimization and forecasting. In this sense, we use the CME that transforms the nonlinear system (2) into a linear system in a feature space.

### 2.2. Kernel conditional mean embedding

A *Reproducing Kernel Hilbert Space* (RKHS) $\mathbb{H}_S$ on $\mathcal{S}$ with kernel $k_S$ is a Hilbert space, satisfying the reproducing property (Schölkopf & Smola, 2002). Let $k_S$ and $k_O$ be positive-definite (pd) kernels on $\mathcal{S}$ and $\mathcal{O}$ with RKHS $\mathbb{H}_S$ and $\mathbb{H}_O$. We denote the kernel-induced features as $\phi_O(o_t) = k_O(o_t, \cdot)$ and $\phi_S(s_t) = k_S(s_t, \cdot)$, referring the $\mathbb{H}_S$ and $\mathbb{H}_O$ as feature spaces respectively.

Kernel mean embedding of distribution $\mathbb{P}_S$ is defined as the expectation of the feature $\phi_S(S) : \mu_{\mathbb{P}_S} = \mathbb{E}_S[\phi_S(S)] \in \mathbb{H}_S$ and always exists for bounded kernels (Fukumizu et al., 2011). With the reproducing property of RKHS, expectations of the function $f$ can easily be computed as $\mathbb{E}_S(f(S)) = \langle \mu_{\mathbb{P}_S}, f \rangle_{\mathbb{H}_S}$ for any $f \in \mathbb{H}_S$. For *characteristic* kernels, these embeddings are injective, uniquely determining the probability distribution (Berlinet & Thomas-Agnan, 2011). In addition to mean embedding, we will need

the (uncentered) cross-covariance operator (Baker, 1973) $\mathcal{C}_{SO} \colon \mathbb{H}_O \to \mathbb{H}_S$, defined as $\mathcal{C}_{SO} = \mathbb{E}_{SO}[\phi_S(S) \otimes \phi_O(O)]$, where $\otimes$ denotes the tensor product. This operator always exists for bounded kernels and can also be viewed as the embedding of the joint distribution $\mathbb{P}_{SO}$, represented as an element in the tensor product space $\mathbb{H}_S \otimes \mathbb{H}_O$. Similarly, the covariance operator $\mathcal{C}_{SS} \colon \mathbb{H}_S \to \mathbb{H}_S$ is defined as $\mathcal{C}_{SS} = \mathbb{E}[\phi_S(s) \otimes \phi_S(s)]$. These operators extend the concepts of covariance matrices from finite-dimensional spaces to infinite kernel feature spaces.

**Conditional mean embedding.** To represent the dynamical and observation models that arise in (1), the conditional mean embedding (CME) plays an important role. For a conditional distribution $\mathbb{P}_{S|o}$ of $S$. The CME of $\mathbb{P}_{S|O}$ is defined as

$$\mu_{\mathbb{P}_{S|o}} = \mathbb{E}_{S|o}[\phi_S(S)|O = o] \quad \text{for any } o \in \mathcal{O}, \quad (3)$$

requiring an operator $\mathcal{C}_{S|O} \colon \mathbb{H}_O \to \mathbb{H}_S$ such that (1) $\mu_{\mathbb{P}_{S|o}} = \mathcal{C}_{S|O}\phi_O(o)$, and (2) $\langle f, \mu_{\mathbb{P}_{S|o}} \rangle_{\mathbb{H}_S} = \mathbb{E}_{S|o}[f(S)|O = o]$ for any $f \in \mathbb{H}_S$. Under standard assumptions[1], the CME operator can be expressed as $\mathcal{C}_{S|O} = \mathcal{C}_{SO}\mathcal{C}_{OO}^{-1}$ (Fukumizu et al., 2004; Song et al., 2009). Given i.i.d samples $\{(s_i, o_i)\}_{i=1}^n \sim \mathbb{P}_{SO}$, an empirical estimate of the CME operator can be obtained as

$$\hat{\mathcal{C}}_{S|O} = \hat{\mathcal{C}}_{SO}(\hat{\mathcal{C}}_{OO} + \lambda I)^{-1} = \Phi_S(K_O + \lambda I)^{-1}\Phi_O^T, \quad (4)$$

where $\lambda$ is the regularization parameter, $K_O$ is the Gram matrix $[K_O]_{ij} = k_O(o_i, o_j)$, and $\Phi_S = [\phi_S(s_1), ..., \phi_S(s_n)]$ and $\Phi_O = [\phi_O(o_1), ..., \phi_O(o_n)]$ are the feature matrices stacked by columns.

## 3. Method

In this section, we introduce our *Tensor-Var* approach, which embeds 4D-Var into the kernel feature space and provides a theoretical analysis demonstrating the existence of linear dynamics with consistent convergence between the original and feature space solutions. To effectively address incomplete observations, we propose an inverse observation operator that leverages consecutive historical observations. Additionally, we propose a method to learn adaptive deep features (DFs) using neural networks within the Tensor-Var framework, improving real-world applicability. Finally, we analyze optimization performance of Tensor-Var compared to existing 4D-Var based methods, demonstrating its efficiency.

### 3.1. CME of 4D-Var in RKHS

Having introduced the necessary tools for manipulating kernel embeddings, we now focus on learning the linearized

---

[1] (1) $\mathcal{C}_{SS}$ is injective, and (2) $\mathbb{E}_{S|o}[f(S)|O = o] \in \mathbb{H}_S$ for any $f \in \mathbb{H}_S$ and $o \in \mathcal{O}$.

models of the system in (1).

**CME of dynamical model.** Let $S^+$ be the one-step forward of $S$. Given the dynamical model $F$ in (1) and the kernel feature $\phi_S$, the CME operator $\mathcal{C}_{S^+|S}$ can be recognized as the best linear approximation in the feature space $\mathbb{H}_S$ that minimizes the regression residual $\mathbb{E}[\|\phi_S(S^+) - \mathcal{C}\phi_S(S)\|^2]$. Given finite data $\{(s_i^+, s_i)\}_{i=1}^N$ sliced from the system trajectory $s_{1:N+1}$, we can obtain the empirical estimate $\hat{\mathcal{C}}_{S^+|S}$ as (4) with theoretical support of convergence (Fukumizu et al., 2013; Klus et al., 2020). The CME operator $\hat{\mathcal{C}}_{S^+|S}$ effectively characterizes the system dynamics as a linear model and simplifies the 4D-Var as a convex optimization in the feature space $\mathbb{H}_S$.

**CME of inverse observation model.** Analogous to the dynamical model, the observation model $G$ in (1) can be linearized by the CME operator $\mathcal{C}_{O|S}$, which has been used as observation models for filtering algorithms (Song et al., 2009; Fukumizu et al., 2013; Kanagawa et al., 2016; Gebhardt et al., 2019). Most approaches assume a complete observation setting, where the observations can fully determine the state. In practice, observations are often incomplete representations of states, with $n_s > n_o$, leading to underdetermined systems (Liu et al., 2022). In such systems, the lack of a bijective mapping between the state and observation spaces means that observations cannot uniquely determine the system's state. As a result, the optimization problem in 4D-Var may produce suboptimal solutions (Asch et al., 2016). It is theoretically challenging to distinguish any two mixtures of states based on a single-step observation if $n_s > n_o$. By introducing the past $m$ consecutive observations as history $h_t = o_{t-m-1:t-1} \in \mathbb{R}^{m \times n_o}$, the joint information from history and current observation is enough to estimate the system state. The choice of history length is critical: Too short lacks sufficient information, while too long is inefficient. Empirically, we performed an ablation study to assess the effects of history length, as detailed in the Subsection 4.4.

To effectively incorporate history, we introduce another kernel feature $\phi_H$ with an induced RKHS $\mathbb{H}_H$. The space $\mathbb{H}_{OH} = \mathbb{H}_O \otimes \mathbb{H}_H$ is called a tensor product RKHS on $O \times H$ with associated kernel feature $\phi_{OH}(o, h) = \phi_O(o) \otimes \phi_H(h)$. As shown in (Song et al., 2013; Muandet et al., 2017), the CME operator can be extended to high-order features, allowing us to embed the joint distributions over $s_t$, $o_t$, and $h_t$ into the feature space (Song et al., 2009; 2013). The tensor product feature $\phi_{OH}$ captures the high-order dependencies between observations and the history. The CME operator $\mathcal{C}_{S|OH}$ is the linear inverse observation model that minimizes the state estimation error given observations and history. Given dataset $\{(s_i, o_i, h_i)\}_{i=1}^N$, the empirical counterpart $\hat{\mathcal{C}}_{S|OH} = \hat{\mathcal{C}}_{SOH}(\hat{\mathcal{C}}_{(OH)(OH)} + \lambda I)^{-1}$ follows

*Table 1.* Comparison of Tensor-Var with other algorithms. Here, $\epsilon$ is the error threshold and $T$ is the length of the assimilation window.

| Methods | Space Transformation | | Non-Space Transformation | |
| --- | --- | --- | --- | --- |
| | Tensor-Var | Latent 4D-Var | 4D-Var (w/o or w/ Adjoint) | Frerix et al. (2021) |
| Convexity | ✓ | ✗ | ✗ | ✗ |
| Convergence Rate | Linear | Sublinear | Sublinear | Sublinear |
| Iteration Time | $\mathcal{O}\left(\log\left(1/\epsilon\right)\right)$ | $\mathcal{O}(1/\epsilon^2)$ | $\mathcal{O}\left(1/\epsilon^2\right)$ | $\mathcal{O}\left(1/\epsilon^2\right)$ |
| Computational Complexity | $\mathcal{O}\left(Td_s\log\left(1/\epsilon\right)\right)$ | $\mathcal{O}\left((Td_s)^2/\epsilon^2\right)$ | w/o Adjoint: $\mathcal{O}\left((Tn_s)^2/\epsilon^2\right)$
w/ Adjoint: $\mathcal{O}\left(Tn_s/\epsilon^2\right)$ | $\mathcal{O}\left((Tn_s)^2/\epsilon^2\right)$ |

the same way as (4). The $\hat{\mathcal{C}}_{SOH}$ is the empirical high order tensor, e.g., $\hat{\mathcal{C}}_{SOH} = \sum_{i=1}^{N}\phi_S(s_i)\otimes\phi_O(o_i)\otimes\phi_H(h_i)$.

**Feature space 4D-Var.** Using the kernel features, we linearize the original nonlinear dynamics and observations in the feature space $\mathbb{H}_S$. This transformation enables us to reformulate the 4D-Var optimization objective (2) into the feature space and optimize over a sequence of elements $z_{0:T}$ in feature space $\mathbb{H}_S$:

$$
\min_{z_{0:T}} \|z_0 - \phi_S(s_0^b)\|_{\mathcal{B}^{-1}}^2 + \sum_{t=0}^{T-1}\|z_{t+1} - \mathcal{C}_{S^+|S}z_t\|_{\mathcal{Q}^{-1}}^2
$$
$$
+ \sum_{t=0}^{T}\|z_t - \mathcal{C}_{S|OH}\phi_{OH}(o_t, h_t)\|_{\mathcal{R}^{-1}}^2,
$$
(5)

where the $\mathcal{B}$, $\mathcal{R}$, and $\mathcal{Q}$ are the covariance operators for the background error, observation error, and model error in the feature space. In this work, we estimate the three operators as the empirical error covariance matrices from the training dataset (the explicit estimation can be found in Appendix D.1). We present the pseudo-algorithms in Appendix D.2, as shown in Algorithms 2 and 3.

### 3.2. Learning the deep features within Tensor-Var

Using the pre-determined kernel features has theoretical guarantees. However, it maps data into an infinite-dimensional feature space, e.g., radial basis function, making estimation challenging due to polynomial scaling with sample size. On the other hand, these feature maps struggle with irregular or high-dimensional data, often resulting in poor performance. Learned deep features (DFs) have emerged as alternatives to generic kernel features (Xu et al., 2022; Kostic et al., 2023; Shimizu et al., 2024), projecting the data into a fixed-dimensional feature space. To improve scalability, we integrate DFs with the Tensor-Var framework and validate their effectiveness through experiments.

**Learning the state feature.** Recall that the CME operator $\mathcal{C}_{S^+|S}$ is the best linear approximation of system dynamics in feature space. We propose to jointly learn the

feature $\phi_{\theta_S}: \mathbb{R}^{n_s} \to \mathbb{R}^{d_s}$ with $\hat{\mathcal{C}}_{S^+|S}$ by minimizing the loss $L(\theta_S) = \min_{\mathcal{C}\in\mathbb{R}^{d_s\times d_s}} L(\mathcal{C}, \theta_S) = \mathbb{E}\big[\|\phi_{\theta_S}(s^+) - \hat{\mathcal{C}}_{S^+|S}\phi_{\theta_S}(s)\|^2\big]$. Predictions using $\hat{\mathcal{C}}_{S^+|S}$ are in the feature space; however, for DA problems, reconstruction to the original state space is required, which is known as the preimage problem (Honeine & Richard, 2011). Here, we learn an inverse feature $\phi_{\theta'_S}^{\dagger}$ to solve the preimage problem during training, avoiding repeated optimization whenever computing preimages. The final training loss is the combination of the two terms as

$$
L(\theta_S, \theta'_S) = \mathbb{E}\big[\|\phi_{\theta_S}(s^+) - \hat{\mathcal{C}}_{S^+|S}\phi_{\theta_S}(s)\|^2\big]
$$
$$
+ w\,\mathbb{E}\Big[\|s - \phi_{\theta'_S}^{\dagger}\big(\phi_{\theta_S}(s)\big)\|^2\Big],
$$

where $w \in (0, 1]$ is a weighting coefficient, and $\hat{\mathcal{C}}_{S^+|S}$ is computed as the CME over training batches. Note that using DFs corresponds to a linear kernel in the learned feature space, where $k_{\theta_S}(s_i, s_j) = \phi_{\theta_S}(s_i)^T\phi_{\theta_S}(s_j)$.

**Learning the observation and history features.** Similar to learning the state feature, $\mathcal{C}_{S|OH}$ is the minimizer of the regression problem mapping the tensor product of observation and history features to the state feature. In this phase, we learn the DFs for observation $\phi_{\theta_O}: \mathbb{R}^{n_o} \to \mathbb{R}^{d_o}$, history $\phi_{\theta_H}: \mathbb{R}^{n_h} \to \mathbb{R}^{d_h}$, and $\mathcal{C}_{S|OH}$ jointly with the loss function:

$$
L(\theta_O, \theta_H) = \mathbb{E}\big[\|\phi_{\theta_S}(s) - \hat{\mathcal{C}}_{S|OH}[\phi_{\theta_O}(o)\otimes\phi_{\theta_H}(h)]\|^2\big],
$$

where $\hat{\mathcal{C}}_{S|OH}$ is computed over training batches in parallel, with the $\otimes$ denoting the Kronecker product in practice.

**Improved 4D-Var Optimization.** Tensor-Var improves 4D-Var optimization by transforming the nonlinear problem into a linear convex optimization, enabling global optimality and faster convergence in feature space. Unlike conventional 4D-Var, which relies on costly Hessian computations and suffers from sublinear convergence, Tensor-Var achieves linear convergence through linearized dynamics $z_{t+1} = \mathcal{C}_{S^+|S}z_t$. Table 1 presents a theoretical comparison of optimization properties between Tensor-Var and existing 4D-Var-based methods, including ML-hybrid 4D-Var

(Frerix et al., 2021), latent 4D-Var (Cheng et al., 2024), and traditional 4D-Var with or without an adjoint model. As shown in Table 1, due to its linear dynamics, its iteration time scales as $\mathcal{O}(\log(1/\epsilon))$ with error threshold $\epsilon$, which is more efficient than $\mathcal{O}(1/\epsilon^2)$ the complexity of most 4D-Var algorithms, as supported by empirical results in Section 4.1. Notably, there is an additional error source from the feature mapping, which is not explicitly discussed here; a more in-depth analysis can be found in Appendix B.

### 3.3. Theoretical analysis.

In Section 3.1, we discuss how the dynamical system $F$ in (1) can be embedded as a linear system in the feature space. However, two important questions remain: *1) Does such a linear dynamical system exist? 2) Are the solutions of the original and feature space 4D-Var consistent?* In this section, we provide affirmative answers to both questions using the theory of Kazantzis-Kravaris/Luenberger (KKL) observers (Andrieu & Praly, 2006). We give a road map of the theoretical analysis with the main results and refer the reader to Appendix C for details.

Under the mild assumptions that (1) the dynamical model $F$ is first-order differentiable and (2) the kernel features are all first-order differentiable, the kernel feature $\phi_S$ satisfies the necessary conditions as the state transformation in the KKL observer framework. This transformation enables us to represent the nonlinear dynamical system as a linear system in a higher-dimensional feature space, as established by KKL observer theory (Tran & Bernard, 2023). This result confirms the existence of such a linear system, thus answering question 1). The KKL observer theory provides a theoretical foundation for our approach, bridging nonlinear dynamics and linear 4D-Var methods (Andrieu & Praly, 2006). A detailed derivation proving that $\phi_S$ satisfies the conditions as the state transformation of the KKL observer can be found in Appendix C.

We consider the nonlinear system in (1) within a compact state space and assume that the cost function in 2 has a unique solution. Given that $\phi_S$ is a state transformation in the KKL observer, the system in the original state space can be represented linearly in the feature space. The solution in the feature space has a consistent convergence to the unique solution of the original 4D-Var problem, minimizing with respect to the cost function 5, answering question 2). A formal theorem with detailed proofs can be found in Theorem C.9 in Appendix C.2.

## 4. Experiment

To evaluate our proposed method, the comparison is conducted on a series of benchmark domains, representing the optimization problem (2) of increasing complexity, includ-

ing (1) Lorenz 96 system (Lorenz, 1996) with $n_s = 40$ and 80. (2) Kuramoto-Sivashinsky (KS) equation: a fourth-order nonlinear PDE system (Papageorgiou & Smyrlis, 1991) with $n_s = 128$ and 256, representing different spatial resolution. For both systems, we use a nonlinear observation model $o = G(s) = 5\arctan(s\pi/10) + \epsilon$, where $\epsilon$ is white noise with a standard deviation of 0.01 times the standard deviation of the state variable distribution, and only 20% states can be observed. To assess the practical applicability of Tensor-Var, we evaluate its performance in global medium-range weather forecasting (i.e. 3-5 days) by using a subset of the ECMWF Reanalysis v5 (ERA5) dataset for training and testing, with further details in Subsection 4.2 and (Rasp et al., 2024). Moreover, we incorporate observation locations extracted from the real-time weather satellite track into the NWP experiment with higher spatial-resolution in Subsection 4.3.

**Baselines.** We compare our method against several baseline approaches: (1) 3D-Var with a known observation model, (2) a model-based 4D-Var algorithm that assumes known dynamical and observation models, (3) a learned inverse observation model with the known dynamical model, as proposed by (Frerix et al., 2021). In the two NWP problems, we include another three baselines: Latent 3D-Var and Latent 4D-Var (Cheng et al., 2024), which perform variational data assimilation in a latent space learned via an autoencoder, with Latent 4D-Var also modeling latent-space dynamics; and Fengwu 4D-Var (Xiao et al., 2024), an AI-embedded 4D-Var framework that integrates an advanced neural weather forecasting model into the DA process. These competitive baselines cover both operational Var-DA and ML-hybrid Var-DA methods.

### 4.1. Evaluation and results.

In each experiment, we measure the quality of assimilation using the Normalized Root Mean Square Error (NRMSE) $\sqrt{\frac{1}{T}\sum_{t=0}^{T}\|\hat{s}_t - s_t\|^2}/(s_{\max} - s_{\min})$ over the assimilation window, where $s_{\max}, s_{\min}$ are the maximum and minimum state values in the training dataset. These results and the average evaluation time for each algorithm are reported in Table 2. All metrics are evaluated 20 times with different initial conditions, reporting the mean and standard deviation. For Tensor-Var, the history length $m$ is selected using a cross-validation approach as an ablation study in Subsection 4.4, and the objective function in the feature space is minimized using quadratic programming, implemented via CVXPY (Diamond & Boyd, 2016). The baselines use the L-BFGS method (Nocedal & Wright, 2006) with 10 history vectors for the Hessian approximation. For 4D-Var baselines, we consider two cases (with and without adjoint models). The background state $s_b$ is set to the mean state of the training set.

*Table 2.* Comparison of DA performances. All baseline methods use the strong-constraint 4D-Var objective, while our approach uses the weak-constraint 4D-Var objective. Evaluation times and iterations are reported as each assimilation window's mean and standard deviation. Our method consistently outperforms the baselines across all benchmark domains.

| Domain | Algorithm | NRMSE (%) | Evaluation time ($10^{-2}$s) | Iteration time |
|---|---|---|---|---|
| Lorenz 96 $n_s = 40$ $n_o = 8$ | 3D-Var | $14.17 \pm 0.93$ | $12.59 \pm \mathbf{0.39}$ | $13 \pm 2$ |
| | 4D-Var | $12.27 \pm 1.41$ | $210.52 \pm 3.87$ | $17 \pm 6$ |
| | 4D-Var by Adjoint | $12.18 \pm 1.46$ | $33.78 \pm 2.11$ | $16 \pm 4$ |
| | Frerix et al. (2021) | $9.89 \pm 1.63$ | $167.43 \pm 1.33$ | $11 \pm 3$ |
| | Ours | $\mathbf{8.32 \pm 0.87}$ | $12.51 \pm 1.97$ | $\mathbf{8 \pm 2}$ |
| Lorenz 96 $n_s = 80$ $n_o = 16$ | 3D-Var | $15.19 \pm 1.09$ | $\mathbf{19.38 \pm 0.37}$ | $14 \pm 2$ |
| | 4D-Var | $12.38 \pm 1.11$ | $322.21 \pm 5.73$ | $22 \pm 8$ |
| | 4D-Var by Adjoint | $12.44 \pm 2.55$ | $48.07 \pm 2.59$ | $22 \pm 8$ |
| | Frerix et al. (2021) | $10.79 \pm \mathbf{0.57}$ | $286.11 \pm 2.43$ | $14 \pm 2$ |
| | Ours | $\mathbf{9.04} \pm 1.32$ | $21.09 \pm 0.79$ | $\mathbf{9 \pm 1}$ |
| Kuramoto-Sivashinsky $n_s = 128$ $n_o = 32$ | 3D-Var | $17.64 \pm 1.27$ | $\mathbf{16.48 \pm 1.17}$ | $\mathbf{13 \pm 0}$ |
| | 4D-Var | $15.46 \pm 1.07$ | $94.83 \pm 3.89$ | $77 \pm 4$ |
| | 4D-Var by Adjoint | $15.43 \pm \mathbf{0.87}$ | $22.14 \pm 4.01$ | $72 \pm 4$ |
| | Frerix et al. (2021) | $10.25 \pm 1.34$ | $63.28 \pm 1.91$ | $28 \pm 1$ |
| | Ours | $\mathbf{9.69} \pm 1.56$ | $19.58 \pm 1.23$ | $18 \pm 2$ |
| Kuramoto-Sivashinsky $n_s = 256$ $n_o = 64$ | 3D-Var | $16.66 \pm 0.69$ | $\mathbf{18.81 \pm 0.92}$ | $\mathbf{15 \pm 1}$ |
| | 4D-Var | $10.67 \pm 0.62$ | $99.68 \pm 2.35$ | $83 \pm 3$ |
| | 4D-Var by Adjoint | $10.23 \pm 0.21$ | $25.77 \pm 2.13$ | $81 \pm 3$ |
| | Frerix et al. (2021) | $8.87 \pm 0.55$ | $71.39 \pm 1.23$ | $31 \pm 2$ |
| | Ours | $\mathbf{4.31 \pm 0.19}$ | $21.37 \pm 1.36$ | $23 \pm 3$ |

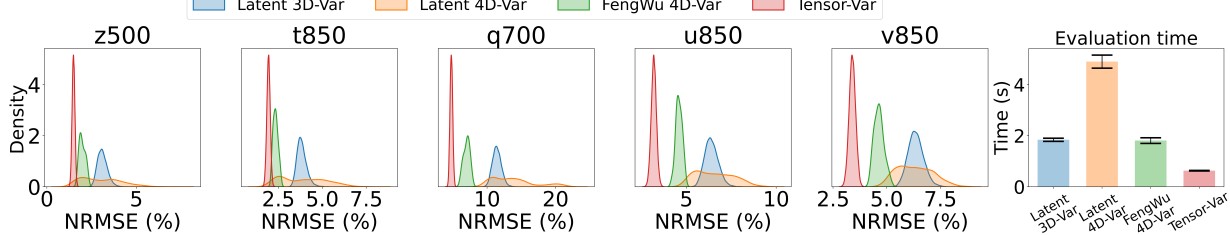

*Figure 2.* Comparison of distribution of NRMSE (%) across different atmospheric variables (z500, t850, q700, u850, v850) for Latent 3D-Var, Latent 4D-Var, Fengwu 4D-Var, and Tensor-Var. The rightmost bar plot shows evaluation times and error bars indicate the standard deviation for evaluation time.

As shown in Table 2, our method consistently outperforms the other baseline methods in all metrics. In all four tasks, Tensor-Var achieves the lowest mean and standard deviation of NRMSE for assimilation accuracy, demonstrating strong generalization from the Lorenz-96 systems to the more complex KS systems. The better performances are attributed to the linearization of the dynamics and the history-augmented inverse observation operator, which makes the 4D-Var optimization convex with more reliable convergence to the global optimum in the feature space. The larger NRMSE errors observed in the model-based 3D-Var and 4D-Var methods are due to the fact that incomplete observations lead to uncontrolled errors in the unobserved state dimensions. Although the ML-hybrid 4D-Var method (Frerix et al., 2021) employs a learned inverse observation model, it struggles with generalization due to the incomplete observations, leading to poor performance on test data. By incorporating historical information, our approach effectively controls estimation errors in the unobserved state dimensions. This

improvement is demonstrated in Figures 7 and 8 in Appendix E.1.2, which qualitatively compares the assimilation performance in the Lorenz-96 and KS systems.

In terms of computational efficiency, the linear structure and convexity of the Tensor-Var enable a linear optimization convergence rate and much lower computational complexity, as shown in Table 1. Our results confirm these improvements by showing lower iteration times and overall evaluation times compared to other baselines. Moreover, the computational cost of feature mapping is negligible relative to the 4D-Var optimization, and the lowest NRMSE validates consistency of the optimization across both the original and feature spaces.

### 4.2. Global NWP

Next, we consider a global NWP problem. The European Centre for Medium-Range Weather Forecasts (ECMWF) Atmospheric Reanalysis (ERA5) dataset (Hersbach et al.,

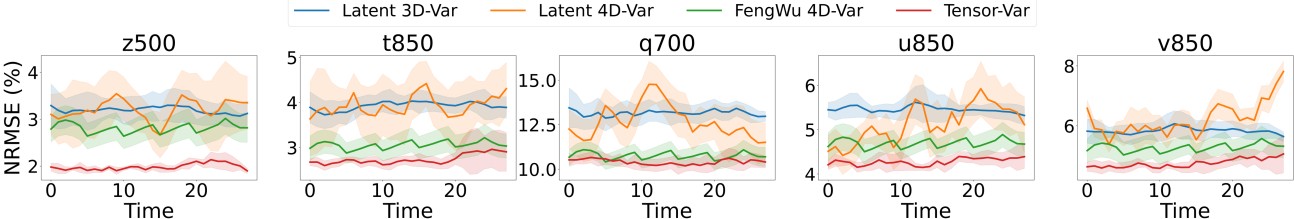

*Figure 3.* Comparison of assimilation NRMSE (%) over a 7-day horizon for five atmospheric variables from Latent 3D-Var, Latent 4D-Var, Fengwu 4D-Var, and Tensor-Var. Each time-step represents a 6-hour interval.

2020) provides the best estimate of the dynamics of the atmosphere covering the period from 1940 to present. The 500hPa geopotential, 850 hPa temperature, 700 hPa humidity, and 850 hPa wind speed (meridional and zonal directions) at $64\times32$ resolution are considered here. The data is sourced from the WeatherBench2 repository (Rasp et al., 2024). Observations are sampled randomly from the grid with a 15% spatial coverage and with additive noise (0.01 times the standard deviation of the state variable) (see Figure 9). Latent dimensions of the two baseline methods are set to match the feature dimension $d_s$ in our approach. The evaluation metric is the area-weighting RMSE over grid points (see more details in Appendix E.3). We trained all models in ERA5 data from 1979-01-01 to 2016-01-01 and tested on data post-2018, with a qualitative evaluation shown for 2018-01-01 00:00 in Figure 9.

Figure 2 (left 1-5) shows the distributions of NRMSE (%) across different atmospheric variables (z500, t850, q700, u850, v850) for Latent 3D-, 4D-Var, and Tensor-Var, evaluated on a test data set consisting of two years of data from 2018-01-01 00:00 to 2020-01-01 00:00. Our approach consistently achieves the lowest mean and standard deviation of NRMSE for all variables, demonstrating improved performance in both assimilation accuracy and robustness. The latent spaces of 3D- and 4D-Var are learned by the autoencoder purely based on the reconstruction loss without considering the system dynamics. This weakens the forecasting abilities of the dynamical systems, introducing extra errors in the 4D-Var optimization compared to the 3D-Var optimization. In contrast, our approach jointly learns feature space representation and linear system dynamics, resulting in more accurate forecasting and improved 4D-Var performances. The rightmost barplot in Figure 2 shows the evaluation times on an Nvidia RTX-4090 GPU. Tensor-Var achieves faster computation than Latent 3D-Var and Latent 4D-Var because its deep features are used only to map data into feature space rather than being directly involved in gradient-based optimization via AD. In addition to the assimilation results, we evaluate the forecast quality of Tensor-Var based on the assimilated state; quantitative results can be found in Appendix E.3 Figure 10.

### 4.3. Assimilation from Satellite Observations

In the final experiment, we consider observations from satellite track locations instead of random observation points in Section 4.2, addressing a more realistic DA problem in global NWP. Weather satellites offer crucial observations, but their dynamic positions and spatial-temporal sparsity make data assimilation more challenging. The spatial resolution of the grid increases to $240 \times 121$ in this case.

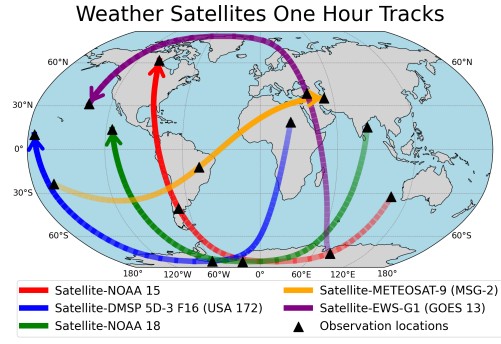

*Figure 4.* Selected satellite tracks over a one-hour horizon, with observations (black triangles) sampled at half-hour intervals.

An example of satellite tracks and observation distribution is shown in Figure 4. We extract satellite track data (latitude and longitude coordinates) from CelesTrak[2] for the same periods as Section 4.2, matching it with ERA5 data to generate practical observations. These observations include satellite locations within two hours before the assimilation time, sampled at half-hour intervals, with an average coverage of approximately 6%, see Figure 5. Other experimental settings, such as data volume, training/testing periods, variables, and the 4D-Var window, align with Section 4.2. In this experiment, we evaluate our proposed method in an online operational scenario, where the DA is continuously applied. To assess the uncertainties, all methods are evaluated 10 times by randomly sampling sequences from the test dataset. We present the mean and standard deviation of the NRMSEs in Figure 3 over a 7-day time horizon. Tensor-Var

---

[2]CelesTrak provides public orbital data for a wide range of satellites, including those with meteorological sensors at `www.celestrak.com`. The data include positional details and temporal information, allowing accurate real-time satellite tracking.

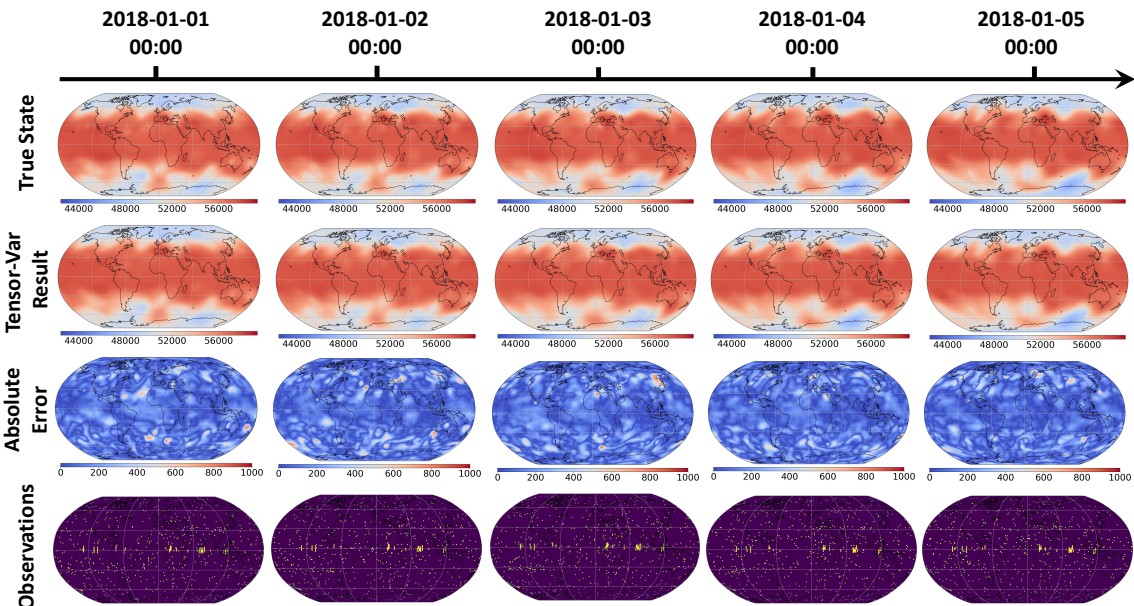

*Figure 5.* Visualization of continuous assimilation results, absolute errors, and observation locations for z500 (geopotential), starting from 2018-01-01 00:00. The observation coverage, defined as the ratio of the number of observations to the number of grid points, is 6.37%.

consistently achieves the lowest mean and standard deviation, demonstrating its robustness in a large-scale system with practical observations. Figure 5 presents the qualitative assimilation results for the variable z500 with observations; results for other variables are provided in Appendix E.4. Dynamic satellite observations impact assimilation accuracy, with clustered observations near the equator from geosynchronous satellites substantially reducing errors in the corresponding regions. The results demonstrate that Tensor-Var excels in accuracy and robustness when handling large-scale systems with practical observations, showing its potential for applications in operational DA and forecasting.

### 4.4. Ablation study

To support our empirical results, we perform ablation studies on the 40- and 80-dimensional Lorenz-96 system to investigate (1) the effect of history length in learning the operator $\hat{\mathcal{C}}_{S|OH}$, and explore (2) the effect of DFs $\phi_S$ with different feature dimensions.

**Effect of history length.** We explore the effect of history length $m$ on learning the inverse observation operator and its impact on state estimation accuracy. According to the theory in (Liu et al., 2022), the history length can be chosen as $m \propto \log(n_s)^3$. The ablation study is conducted by scaling $m$ proportionally to $m \approx C \log n_s$ where the constant $C$ was adjusted. The dimensions $d_s, d_o, d_h$ are fixed

---

[3]Please note that the result omits the class of systems with exponential dependency on the history length.

to be the same as the experiments in Subsection 4.1. Table 3 (right) shows that incorporating history ($C > 0$) significantly improves the accuracy of the state estimation, with the NRMSE decreasing as $C$ increases. However, improvements become marginal when increasing $C$ beyond a certain point (around $C = 4$). This indicates a trade-off, where increasing the history length beyond a certain threshold yields little additional benefit in state estimation.

**Effect of DFs.** This experiment compares DFs with varying dimensions $d_s = 20, 40, 60, 80$ to a Gaussian kernel feature. To scale up the Gaussian kernel, we apply Nyström approximation and kernel PCA (See Appendix E.5 for details). All three features are Gaussian kernels. The preimage of the system state is learned using kernel ridge regression on the low-dimensional representations. As shown in Table 3 (left), Gaussian kernel performance is close to the best DFs in $n_s = 40$ but degrades in $n_s = 80$. In a lower-dimensional problem, the Gaussian kernel features are more robust than DFs, but their performance can degrade with increasing system dimensionality and complexity. DFs with $d_s = 60$ and 120 perform consistently well, whereas $d_s = 20$ and 40 underperform, indicating trade-offs between data size, model complexity, and system dimension.

## 5. Conclusion

In this paper, we propose Tensor-Var, a framework that uses kernel conditional mean embedding to linearize nonlinear models in 4D-Var, making optimization tractable and

*Table 3.* Comparison of different features and history lengths with NRMSE (%) as the metric for state estimation accuracy. GK denotes the Gaussian kernel.

| Feature Dimension (NRMSE %) | | | History Length (NRMSE %) | | |
| --- | --- | --- | --- | --- | --- |
| $d_s$ | $n_s = 40$ | $n_s = 80$ | $C$ | $n_s = 40$ | $n_s = 80$ |
| 20 | $16.7 \pm 2.1$ | $17.3 \pm 2.6$ | 0 | $11.7 \pm 3.4$ | $10.8 \pm 2.6$ |
| 40 | $14.1 \pm 1.3$ | $16.8 \pm 2.2$ | 1 | $9.3 \pm 2.8$ | $8.2 \pm 1.7$ |
| 60 | $\mathbf{8.3} \pm 0.9$ | $11.4 \pm 0.9$ | 2 | $7.7 \pm 1.4$ | $7.4 \pm 0.9$ |
| 120 | $9.7 \pm 0.7$ | $\mathbf{10.9 \pm 0.9}$ | 4 | $7.7 \pm 0.9$ | $7.1 \pm 0.9$ |
| GK | $8.4 \pm \mathbf{0.5}$ | $11.7 \pm 1.4$ | 8 | $\mathbf{7.5 \pm 0.5}$ | $\mathbf{7.0 \pm 0.8}$ |

efficient. By learning adaptive deep features, Tensor-Var addresses the scalability typically associated with traditional kernel methods. Our inverse observation operator, which incorporates historical observations, improves accuracy and robustness with incomplete observations. Experiments on two chaotic systems and global weather forecasting show that Tensor-Var outperforms state-of-the-art hybrid ML-DA models in both accuracy and efficiency.

**Limitations and future work.** Tensor-Var requires access to system states to learn dynamics and observation models, which may not be feasible in practice. Future work will focus on learning these models directly from observations and calibrating dynamics in feature space. Additionally, the simplified error covariance used in 4D-Var may underestimate system correlations; refining these within Tensor-Var could improve assimilation and performance.

## Acknowledgments

The authors thank the reviewers and the ICML 2025 program committee for their insightful feedback and constructive suggestions that have helped improve this work. We also acknowledge the support from the University College London (Dean's Prize, Chadwick Scholarship), the Engineering and Physical Sciences Research Council projects (EP/W007762/1), the United Kingdom Atomic Energy Authority (NEPTUNE 2057701-TN-03), and the Royal Academy of Engineering (IF-2425-19-AI165).

## Impact Statement

This paper aims to advance machine learning applications in studying the long-term properties of large-scale chaotic systems. There are many potential societal consequences of our work, none of which we feel must be specifically highlighted here.

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

## A. Table of Notations

| Notations | Meaning |
|:---:|:---:|
| $S, O, H$ | random variables of state, observation, and history. |
| $s, o, h$ | samples of state, observation, and history. |
| $\mathcal{S}$ | domain of state space. |
| $\mathcal{O}$ | domain of observation space. |
| $n$ | dimension of original spaces with specified subscripts. |
| $\mathbb{P}$ | probability distribution with specified subscripts. |
| $\phi_S(s) = k_S(s, \cdot)$ | kernel feature map for state. |
| $\phi_O(o) = k_O(o, \cdot)$ | kernel feature map for observation. |
| $\phi_H(h) = k_H(h, \cdot)$ | kernel feature map for history. |
| $\mathbb{H}_S, \mathbb{H}_O, \mathbb{H}_H$ | induced kernel feature space for $S, O$ and $H$. |
| $z$ | indicate the elements in feature space. |
| $\mathcal{C}$ | Conditional mean embedding (CME) operators with specified subscripts. |
| $B, R, Q$ | error matrices for background, observation, and dynamical model in original space. |
| $\mathcal{B}, \mathcal{R}, \mathcal{Q}$ | error matrices for background, observation, and dynamical model in feature space. |
| $\phi_{\theta_S}$ | deep feature for state with neural network parameters $\theta_S$. |
| $\phi^{\dagger}_{\theta'_S}$ | inverse feature for state with neural network parameters $\theta_S$. |
| $\phi_{\theta_O}$ | deep feature for state with neural network parameters $\theta_O$. |
| $\phi_{\theta_H}$ | deep feature for state with neural network parameters $\theta_H$. |
| $d$ | dimension of feature spaces with specified subscripts. |

## B. Variational Data Assimilation Optimization

A sequence $\{x_k\}$ generated by an optimization algorithm converges **linearly** to $x^*$ if there exists $C \in (0,1)$ such that $\|x_{k+1} - x^*\| \leq C\|x_k - x^*\|$, implying $\|x_k - x^*\| = \mathcal{O}(C^k)$. Convergence is **sublinear** if $\|x_k - x^*\| \to 0$ as $k \to \infty$ but slower than linear, often expressed as $\|x_k - x^*\| = \mathcal{O}(1/k^\alpha)$ for $\alpha > 0$. **Superlinear** convergence occurs when $\lim_{k \to \infty} \|x_{k+1} - x^*\|/\|x_k - x^*\| = 0$, with **quadratic convergence** as a special case where $\|x_{k+1} - x^*\| \leq C\|x_k - x^*\|^2$.

### B.1. Analysis of Optimization

**Tensor-Var.** Our approach based on CME is theoretically linear in the learned feature space. Under this condition, it formulates a convex quadratic programming (QP) problem with linear dynamic constraints, as illustrated in Equation (5). From the convergence rates of the Newton optimization method (Wright, 2006), the algorithm exhibits a linear convergence rate in worst-case and a superlinear convergence rate in best-case. Consequently, the number of iterations required to achieve an $\epsilon$-error ranges between $\mathcal{O}(\log(1/\epsilon))$ and $\mathcal{O}(\log\log(1/\epsilon))$. In terms of computational complexity, the QP problem features an invariant Hessian matrix. This invariance allows us to compute the Hessian matrix only once. Incorporating the iteration complexity, the total computational complexity of Tensor-Var is efficiently bounded between $\mathcal{O}(Td_s \log(1/\epsilon))$ and $\mathcal{O}(Td_s \log\log(1/\epsilon))$. This demonstrates that Tensor-Var achieves efficient computation by minimizing redundant Hessian calculations and leveraging the convergence properties of the Newton method. It should be noted that the optimization error in Tensor-Var is in the feature space, not the state space. The true error is quantified as $(\epsilon + \hat{\text{err}})$ it decoding back, where $\hat{\text{err}}$ represents the reconstruction error.

**Latent 4D-Var.** Unlike our linearized CME model, the latent 4D-Var approach (Cheng et al., 2024) directly constructs a neural network-based surrogate model using an autoencoder, thereby no longer depending on computationally intensive numerical models. In this framework, the inherent nonlinearity and non-convexity of the optimization problem remain unchanged. According to optimization theory, the L-BFGS algorithm achieves a worst-case sublinear convergence rate and a best-case superlinear convergence rate (Boyd & Vandenberghe, 2004; Wright, 2006). The best computational complexity can only be achieved only if the problem exhibits strong local convexity around the initial point. Consequently, the number of iterations required to achieve an $\epsilon$-error ranges between $\mathcal{O}(1/\epsilon^2)$ and $\mathcal{O}(\log\log(1/\epsilon))$. Regarding computational complexity, latent 4D-Var does not decompose the quadratic programming problem into $T$ separate subproblems. Instead, it concatenates the $T$ features into a high-dimensional vector of size $Td_s$. At each time step, the Hessian matrix is updated using the BFGS algorithm. Considering the complexity of the iteration, the total computational complexity ranges from $\mathcal{O}((Td_s)^2/\epsilon^2)$ to $\mathcal{O}((Td_s)^2 \log\log(1/\epsilon))$ which makes it computationally expensive. Since optimization is performed in the latent feature space, the true error is quantified as $(\epsilon + \hat{\text{err}})$ upon decoding, where $\hat{\text{err}}$ represents the reconstruction error.

**4D-Var.** The original 4D-Var, when solved using the Newton method, exhibits a similar convergence rate, ranging from sublinear to superlinear. However, the number of iterations scales with the state dimension $n_s$ rather than the latent space dimension $d_s$, leading to a total complexity of $\mathcal{O}((Tn_s)^2/\epsilon^2)$ or $\mathcal{O}((Tn_s)^2 \log\log(1/\epsilon))$ to achieve an $\epsilon$-error. When the adjoint model is available, the total computational complexities can be further reduced to $\mathcal{O}(Tn_s/\epsilon^2)$ or $\mathcal{O}(Tn_s \log\log(1/\epsilon))$ through forward and adjoint computations, each requiring $\mathcal{O}(n_s)$ (Givoli, 2021; Wright, 2006). Since the 4D-Var is directly solved on the original space, there is no reconstruction error that needs to be counted.

**Frerix et al., 2021.** Frerix et al. (2021) propose learning an approximate inverse observation operator using deep learning and transform this problem into the physical state space. The 4D-Var problem is then solved using a standard Newton solver, resulting in convergence properties and computational complexities similar to those of the standard 4D-Var without an adjoint model such as $\mathcal{O}((Tn_s)^2/\epsilon^2)$ or $\mathcal{O}((Tn_s)^2 \log\log(1/\epsilon))$. In practice, (Frerix et al., 2021) argues that solving 4D-Var in a physical state space with an inverse observation operator leads to better initialization, as inverse mapping provides an improved starting point for optimization. This improved initialization results in a better-conditioned optimization landscape, i.e., strongly local convex and achieves a superlinear convergence rate. Our numerical experiments further confirm that their method achieves faster convergence with fewer iterations compared to the standard 4D-Var model (with or without an adjoint model), as shown in Table 2.

A comprehensive summary of these analyses and findings is provided in Table 4.

*Table 4.* Full comparison of Tensor-Var with other algorithms. Here, $\epsilon$ is the error threshold and $T$ is the length of the assimilation window. The best computational complexity can only be achieved only if the problem exhibits strong local convexity around the initial point.

| Methods | Space Transformation | | Non-Space Transformation | |
|---|---|---|---|---|
| | Tensor-Var | Latent 4D-Var | 4D-Var (w/o or w/ Adjoint) | Frerix et al. (2021) |
| Model-based | ✗ | ✗ | ✓ | ✓ |
| Convexity | ✓ | ✗ | ✗ | ✗ |
| Convergence Rate | linear to superlinear | sublinear to linear | sublinear to linear | sublinear to linear |
| Iteration Time | $\mathcal{O}\left(\log\left(1/\epsilon\right)\right)$ | $\mathcal{O}(1/\epsilon^2)$ | $\mathcal{O}\left(1/\epsilon^2\right)$ | $\mathcal{O}\left(1/\epsilon^2\right)$ |
| Worst Computational Complexity | $\mathcal{O}\left(Td_s\log\left(1/\epsilon\right)\right)$ | $\mathcal{O}\left((Td_s)^2/\epsilon^2\right)$ | w/o Adjoint: $\mathcal{O}\left((Tn_s)^2/\epsilon^2\right)$ 
 w/ Adjoint: $\mathcal{O}\left(Tn_s/\epsilon^2\right)$ | $\mathcal{O}\left((Tn_s)^2/\epsilon^2\right)$ |
| Best Computational Complexity | $\mathcal{O}\left(Td_s\log\log\left(1/\epsilon\right)\right)$ | $\mathcal{O}\left((Td_s)^2\log\log(1/\epsilon)\right)$ | w/o Adjoint: $\mathcal{O}\left((Tn_s)^2\log\log\left(1/\epsilon\right)\right)$ 
 w/ Adjoint: $\mathcal{O}\left(Tn_s\log\log\left(1/\epsilon\right)\right)$ | $\mathcal{O}\left((Tn_s)^2\log\log\left(1/\epsilon\right)\right)$ |

# C. Theoretical Analysis

In this section, we provide a theoretical convergence analysis of Tensor-Var, drawing on concepts from control theory and contraction analysis. We begin by introducing comparison functions, including class $\mathcal{K}$ and $\mathcal{KL}$ functions, as well as Lyapunov functions. Using these tools, we examine the convergence of Tensor-Var through a differential equation, demonstrating monotonic contraction based on the Lyapunov direct method. Furthermore, by using comparison functions, we show that contraction in the feature space implies contraction in the original space.

**Assumption 1.** To perform a formal convergence analysis of Tensor-Var, we make a mild assumption about the first-order differentiability of the dynamical system $F$ in (1) with the time derivative $\dot{s} = f(s)$, a standard assumption in convergence studies (Sastry, 2013).

**Assumption 2.** We require that the kernel possess a well-defined first-order derivative, as the convergence analysis is performed in the feature function space. This assumption is common in kernel methods and is satisfied by many widely-used kernels, such as the Gaussian, Fourier, Matérn, and Laplace kernels (Berg et al., 1984; Schölkopf & Smola, 2002).

## C.1. Notations and Technical Lemmas

**Definition C.1** (Class $\mathcal{K}$ function (Gajic & Qureshi, 2008).)**.** A continuous function $\alpha : [0, a] \to [0, \infty)$ is said to belong to class $\mathcal{K}$ if it is strictly increasing and $\alpha(0) = 0$. It is of class $\mathcal{K}_\infty$ if $\alpha(\infty) = \infty$ and $\alpha(r) \to \infty$ as $r \to \infty$.

**Definition C.2** (Class $\mathcal{KL}$ function (Gajic & Qureshi, 2008).)**.** A continuous function $\beta : [0, a] \times [0, \infty) \to [0, \infty)$ is said to belong to class $\mathcal{KL}$ if for each fixed $t$, the mapping $\beta(r, t)$ belonging to class $\mathcal{K}$ with respect to $r$ and, for each fixed $a$ the mapping $\beta(a, t)$ is decreasing with respect to $t$ and $\beta(a, t) \to 0$ as $t \to \infty$.

The $\mathcal{K}$ and $\mathcal{KL}$ are two classes of comparison functions, and we can use the comparison function to analyze the monotone contraction in both spatial and temporal horizons.

**Definition C.3** (Lyapunov Stability (Gajic & Qureshi, 2008))**.** If the Lyapunov function $V$ is globally positive definite, radially unbounded, the equilibrium is isolated and the time derivative of the Lyapunov function is globally negative definite:

$$\frac{dV}{dt}(x) < 0, \qquad \forall \mathbb{R}^n \setminus \{0\}, \tag{6}$$

then the equilibrium is proven to be globally asymptotically stable. The Lyapunov function is a class $\mathcal{K}$ function, which satisfies the condition as follows

$$\alpha_1(\|x\|) \le V(x) \le \alpha_2(\|x\|), \quad , \forall x \in [0, \infty). \tag{7}$$

**Lemma C.4** (Hurwitz stability criterion (Parks, 1962)). *A square matrix $\mathcal{A} \in \mathbb{R}^{n \times n}$ is said to be Hurwitz stable if all the eigenvalues of $\mathcal{A}$ have strictly negative real parts, i.e., for every eigenvalue $\lambda$ of $\mathcal{A}$,*

$$Re(\lambda) < 0. \tag{8}$$

*In other words, the real part of each eigenvalue of the matrix must lie in the left half of the complex plane.*

**Lemma C.5** (Hurwitz stability criterion via Lyapunov function (Sastry, 2013)). *Give a candidate Lyapunov function for linear dynamics as*

$$V(x) = x^T P x, \quad \frac{dx}{dt} = \mathcal{A}x \tag{9}$$

*where $P$ is symmetric, positive definite matrix; $\mathcal{A}$ governs the evolution of dynamics. For the system to be stable, the time derivative $\frac{dV}{dt}$ must be negative definite, i.e., $\frac{dV}{dt} < 0$ for all $x \neq 0$. This means that:*

$$\frac{dV}{dt}(x) = \frac{d}{dt}(x^T P x) = x^T (\mathcal{A}^T P + P\mathcal{A})x < 0 \tag{10}$$

*with*

$$\mathcal{A}^T P + P\mathcal{A} < 0, \tag{11}$$

*where $(\mathcal{A}^T P + P\mathcal{A})$ is negative definite.*

Based on the convergence analysis and Lyapunov theory, a more generalized concept – *contraction metric* is needed to support our paper.

**Definition C.6** (Contraction (Manchester & Slotine, 2017)). *Given the system $\frac{dx}{dt} = f(x, t)$, if there exists a uniformly bounded metric $M(x, t)$ (positive definite) such that*

$$\frac{dM}{dt} + \frac{\partial f}{\partial x}(x, t)^T M + M \frac{\partial f}{\partial x}(x, t) < -cM, \quad c > 0, \tag{12}$$

*then we call the system contracting, and $M(x, t)$ is a contraction metric.*

### C.2. Proof of Key Theorems

To adopt the convergence analysis of the DA problem, we give a mild assumption on the smoothness of dynamics, the first-order derivative exists in the system 1.

**Theorem C.7** (Embedding and Consistent convergence). *Here, we consider system $s_{t+\Delta t} = F_{\Delta t}(s_t) = s_t + \int_t^{t+\Delta t} f(s_\tau)d\tau$ in (1) is first order differentiable with derivative $\frac{ds_t}{dt} = f(s_t)$ for $s \in \mathbb{R}^{n_s}$ with a unique equilibrium point as $s_*$. If there exists a embedding as $\phi_S(s) := [\phi_S^1(s_t), \ldots, \phi_S^{d_s}(s_t)]^T$ with that $d_s \in \mathbb{N} \cup \infty$ satisfying the following properties:*

- ***a.** (embedding) For a finite $d_s$, the $\frac{\partial \phi}{\partial s}(s_t)$ is full-column rank; when $d_s$ is infinite, it is assumed to be rank-$d_s$ countably infinite, i.e. $\{\nabla \phi_S(s_t)\}$ is full-column rank with $\nabla \phi_S(s_t) = [\frac{\partial \phi_S^1}{\partial s}(s_t), \ldots, \frac{\partial \phi_S^n}{\partial s}(s_t), \ldots]^T$.*

- ***b.** (convergence) There exists Hurwitz matrix $\mathcal{A}$ verifying*

$$\frac{d\phi_S}{dt}(s_t) = \mathcal{A}\phi_S(s_t). \tag{13}$$

*Then, the equilibrium $s_*$ and $\phi_S(s_*)$ are global asymptotic convergence.*

*Proof.*

(Embedding.) The embedding property follows from RKHS theory. Since we restrict our analysis to a separable Hilbert space, it has a countable basis either finite or infinite. Thus, the dimension of the RKHS can be infinite, but the rank of the embedding is determined by a countable set of basis functions (Schölkopf & Smola, 2002).

(convergence.) According to the differential equation in Hilbert space, we have

$$
\begin{aligned}
&\frac{d\phi_S}{dt}(s_t) \\
=&\frac{\partial\phi_S}{\partial s_t}(s_t) \cdot \frac{ds_t}{dt} \\
=&\nabla\phi_S(s_t)f(s_t) \\
=&\mathcal{A}\phi_S(s_t).
\end{aligned}
\tag{14}
$$

The second line of (14) follows from the chain rule, the final line represents the time derivative in Hilbert space, where $\mathcal{A}$ is a linear operator that governs the dynamics. Following the work from (Romanoff, 1947; Bobrowski, 2016; Cheng et al., 2023b), we define $\mathcal{A}$ as:

$$
\mathcal{A} := \lim_{t\to 0^+} \frac{\mathcal{C}_{S^+|S} - Id}{t},
\tag{15}
$$

where $\mathcal{C}_{S^+|S}$ is the conditional covariance operator between future and current states in the RKHS. Given the smoothness of the kernel function and the differentiability of the system dynamics, the linear operator $\mathcal{A}$ exists and is well-defined in this context.

Since $s_*$ is the equilibrium point, we can derive a natural result that $f(s_*) = 0$. Invoking (14) (third line) we have $\frac{d\phi_S}{dt}(s_*) = 0$. Thus $\phi_S(s_*)$ is also a local equilibrium point in RKHS. From the embedding property of RKHS, we have a local injective map $\phi : \mathbb{R}^{n_s} \to \mathbb{R}^{d_s}$, which ensures that the convergence properties of the system in the original space are preserved in the feature space. For the neighbourhood around the equilibrium point $s_*$, there exist class $\mathcal{K}$ functions $\alpha_1$ and $\alpha_2$ as

$$
\alpha_1(\|s_t - s_*\|) \leq \|\phi(s_t) - \phi(s_*)\|_2 \leq \alpha_2(\|s_t - s_*\|), \quad \forall s_t \in B(s_*, \epsilon).
\tag{16}
$$

$B(s_*, \epsilon)$ is denoted as $\epsilon-$ball centred at $s_*$. The smoothness of the kernel function and regularity of the dynamics ensure that the system remains well-behaved in feature space, and the convergence properties of the original system carry over to the feature space.

Thus, when the system is locally stable in the original space, the corresponding system in Hilbert space is also locally stable. According to the Hurwitz stability criterion in C.4, the linear operator $\mathcal{A}$ has only negative real part in its eigenvalues, guaranteeing exponential convergence. If $s_*$ is global equilibrium, then $\phi_S(s_*)$ is also a global equilibrium in feature space.

*Remark* C.8. Theorem C.7 (a) implies the existence of a global coordinate. In many situations, when the embedding space is chosen properly, we can have a stronger result that the existence of left inverse such that $\phi^\dagger(\phi(s)) = s$. This result can be naturally connected to the Kazantzis Kravaris/Luenberger (KKL) observers (Tran & Bernard, 2023). The embedding corresponds to the injectivity of the state. Theorem C.7 (b) corresponds to the convergence in the KKL observer. When the embedding space is uniformly injective, the dynamics in feature space become rectifiable dynamics, yielding a stable trajectory if the original system is stable.

Before entering the last main theorem, we need to introduce the KKL observer. Based on the theory of the KKL observer, we will link the convergence problem in feature space with the KKL observer. Please note that we assume that the state space $\mathcal{S}$ is a bounded set and system (1) is forward complete within this bounded set $\mathcal{S}$.

Consider the nonlinear dynamical system in (1) with time derivative as

$$
\frac{ds_t}{dt} = f(s_t); \qquad o_t = G(s_t).
\tag{17}
$$

The design of the KKL observer is as follows:

- Find an embedding map $\mathcal{T} : \mathcal{S} \to \mathbb{R}^{d_s}$ that transforms (17) to new coordinates $\mathcal{T}(s)$ as

$$
\frac{\partial\mathcal{T}}{\partial s}(s)f(s) = \mathcal{A}\mathcal{T}(s) + BG(s),
\tag{18}
$$

where $\mathcal{A} \in \mathbb{R}^{d_s \times d_s}$ is Hurwitz matrix and $B \in \mathcal{R}^{d_s \times n_o}$, such that the system $(\mathcal{A}, B)$ is controllable[4].

---

[4] $R = [\mathcal{A}, \mathcal{A}B, \mathcal{A}^2B, ..., \mathcal{A}^{d_s-1}B]$ has full row rank (i.e. $\text{rank}(R) = d_s$)

- Since $\mathcal{T}$ is injective, the left inverse $\mathcal{T}^\dagger$ exists, i.e., $\mathcal{T}^\dagger(\mathcal{T}(s)) = s$. The KKL observer is then given by

$$\hat{s} = \mathcal{T}^\dagger(\mathcal{T}(\hat{s})). \tag{19}$$

There are certain conditions (Andrieu & Praly, 2006) that (17) needs to satisfy in order to ensure the existence of a KKL observer (18) in the sense that $\lim_{t \to \infty} \|\hat{s}_t - s_t\| = 0$. On the other hand, the map $\mathcal{T} : S \to \mathbb{R}^{d_s}$ need to be *uniformly injective* if there exists a class $\mathcal{K}$ function $\alpha$ in C.1 such that, for every $s_1, s_2 \in S$, satisfying $\|s_1 - s_2\| \leq \alpha(\mathcal{T}(s_1) - \mathcal{T}(s_2))$. Our embedding properties and convergence conditions (as shown in C.7) are satisfied by the two conditions, thus we can assert the existence of a KKL observer.

In this paper, we state the existence of a global linear dynamical system in feature space. We provide a theoretical guarantee that the embedding property of $\phi_S$ can derive the equivalent convergence in the feature space. However, there are two parts that have not been proven: *1). Does the global linear dynamical system exist? 2) Is the embedding space in C.7 properly defined?* We consider the nonlinear system in (1) within a compact space $S$ and the cost function in 2 has a unique solution. According to the condition of KKL observer, we guarantee (1) the existence of such linear system and (2) the solutions in the original space and feature space has consistent convergence properties, with respect to the cost functions (2) and 5, and convergent exponentially to the unique solution.

**Theorem C.9.** *Let $S$ be a bounded set in $\mathbb{R}^{n_s}$. If there exists a $C^1$ function $\mathcal{T} : S \to \mathbb{R}^{d_s}$ which satisfies the following two conditions:*

- *$\mathcal{T}$ is solution of the partial differential equation*

$$\frac{\partial \mathcal{T}}{\partial s}(s) f(s) = \mathcal{A}\mathcal{T}(s) + BG(s), \quad \forall s \in S, \tag{20}$$

*where $\mathcal{A}$ is Hurwitz matrix, and $(\mathcal{A}, B)$ is controllable;*

- *There exists a Lipschitz constant such that for all $(s_1, s_2)$ in $S \times S$, the following inequality holds:*

$$|s_1 - s_2| \leq L|\mathcal{T}(s_1) - \mathcal{T}(s_2)|; \tag{21}$$

*then there exists a continuous function $\mathcal{T}^\dagger : \mathbb{R}^{n_s} \to S$ such for all $(s, \mathcal{T}(s)) \in S \times \mathbb{R}^{n_s}$*

$$\|\mathcal{T}^\dagger(\mathcal{T}(\hat{s}_t)) - s_t\| \leq cL\|\mathcal{T}(\hat{s}_0) - \mathcal{T}(s_0)\| \exp(\sigma_{max}(\mathcal{A})t), \forall t \in [0, \infty), \tag{22}$$

*and where $(s_t, \mathcal{T}(s_t))$ is the solution of system in (1) and (17) at time $t$; $\sigma_{max}(\mathcal{A})$ is the largest eigenvalue of matrix $\mathcal{A}$.*

*Proof.*

It is possible to defined the left inverse function $\mathcal{T}^\dagger : \mathbb{R}^{d_s} \to S$ and this one satisfies,

$$\|s_1 - s_2\| \leq L\|\mathcal{T}(s_1) - \mathcal{T}(s_2)\|, \quad \forall(s_1, s_2) \in S \times S. \tag{23}$$

It yields that the function $\mathcal{T}^{-1} : \mathbb{R}^{d_s} \to S$ is global Lipschitz. Hence, the function $\mathcal{T}^\dagger : \mathbb{R}^{d_s} \to S$ in our problem is a Lipschitz extension on the set $S$ of this function. For more convenience, we denoted $z := \mathcal{T}(s)$. Following the approach - the Mc-Shane formula in (McShane, 1934), we select the $\mathcal{T}^\dagger$ as the function defined by

$$\mathcal{T}^\dagger(w) \in \inf_z \left\{ (\mathcal{T}^{-1}(z) + L\|z - w\|) \right\}. \tag{24}$$

The function is such that for all $s \in S$,

$$\mathcal{T}^\dagger(\mathcal{T}(s)) = s, \tag{25}$$

and for all $(w, s) \in \mathbb{R}^{d_s} \times S$,

$$\|\mathcal{T}^\dagger(w) - s\| \leq \sqrt{d_s}L\|w - \mathcal{T}(s)\|. \tag{26}$$

This implies that along the trajectory $(s_t, z_t)$ of the system satisfying the following result

$$\|\mathcal{T}^\dagger(z_t) - s_t\| \leq \sqrt{d_s}L\|z_t - \mathcal{T}(s_t)\|, \quad \forall t \in [0, \infty). \tag{27}$$

On the another hand, the function $\mathcal{T}$ is solution of the partial differential equation in (18), consequently, this implies that along the trajectory of system $(z_t, s_t) \in \mathbb{R}^{d_s} \times \mathcal{S}$, we have

$$z_t - \mathcal{T}(s_t) = \exp(\mathcal{A}t)(z_0 - \mathcal{T}(s_0)) \quad \forall t \in [0, \infty). \tag{28}$$

Note, since $\mathcal{A}$ is Hurwitz matrix, with $\sigma_{max}(\mathcal{A}) < 0$, it can derive that equation (22) holds and concludes the proof of the theorem.

*Remark* C.10. The KKL observer asserts that the linear representation of the nonlinear system. After establishing the KKL observer, theorem C.9 asserts the convergence of the estimated trajectory to the equilibrium trajectory. Since the embedding property, we can derive that the existence of left inverse based on the Mc-Shane formula in (24). Meanwhile, the convergence holds when pulling back the state to the original space $\mathcal{S}$. Thus we can assert the feature $\phi_S$ in our framework aligns the KKL observer, and the coordinate transformation $\mathcal{T}$ is just our feature $\phi_S$.

# D. Estimation of error matrices and Pseudo Algorithm

### D.1. Estimation of error matrices.

To estimate these error covariance operators in the feature space, we empirically estimate these error matrices in the feature spaces from the training dataset (Gejadze et al., 2008). For $\mathcal{R}$ and $\mathcal{Q}$, we estimated the covariances as $\mathcal{R} = \frac{1}{N} \sum_{i=1}^{N} r_i \otimes r_i$ and $\mathcal{Q} = \frac{1}{N} \sum_{i=1}^{N} q_i \otimes q_i$, where $r_i = \phi_S(s_i) - \hat{\mathcal{C}}_{S|OH} \phi_{OH}(o_i, h_i)$ and $q_i = \phi_S(s_i^+) - \hat{\mathcal{C}}_{S^+|S} \phi_S(s_i)$ are the regression residuals, quantifying the errors of the empirical operators $\hat{\mathcal{C}}_{S|OH}$ and $\hat{\mathcal{C}}_{S^+|S}$. Similarly, we compute the background covariance as the empirical variance over an average of $\{\phi_S(s_i)\}_{i=1}^{N}$[5] This error should decay monotonically over time and stabilize after a sufficiently long time horizon. This is strongly related to the covariance estimation in Kalman filtering (see Chapter 6.7 (Asch et al., 2016)). We leave the investigation of such design for future efforts.

### D.2. Pseudo algorithm.

In this section, we provide the pseudo-algorithm for training Tensor-Var with traditional kernel features in algorithm 2 and training with deep features in algorithm 1. The kernel feature map $\phi : \mathcal{S} \to \mathbb{H}_S$ such that $k(s_i, s_j) = \langle \phi(s_i), \phi(s_j) \rangle_{\mathbb{H}_S}$ may not necessarily have an explicit form (e.g., RBF and Matérn kernels), as long as the $\langle \cdot, \cdot \rangle_{\mathbb{H}_S}$ is an valid inner product. For clarity, we use the polynomial kernel with degree two and constant $c$ as an example:

- Explicit form, $\phi(s) = k(s, \cdot) = (s_1^2, ..., s_{n_s}^2, \sqrt{2}s_1 s_2, ..., \sqrt{2}s_{n_s-1} s_{n_s}, c)$

- Inner product, $k(s_i, s_j) = \langle \phi(s_i), \phi(s_j) \rangle = (s_i^T s_j + c)^2$

Algorithm 3 outlines the procedure of performing data assimilation with trained models.

---

[5]For a cyclic application of Tensor-Var, a better design for $\mathcal{B}$ should be time-dependent, reflecting the error between the estimated system state and the true state, e.g. (Paulin et al., 2022).

---

**Algorithm 1** Tensor-Var training with deep feature

---

**Require:** Data $\mathcal{D} = \{s_i, o_i, h_i, s_i^+\}_{i=1}^N$; Initialized deep features $\phi_{\theta_S}, \phi_{\theta_O}, \phi_{\theta_H}$; the inverse feature $\phi_{\theta_S'}^\dagger$ training epoch $K$, learning rate $\alpha$, batch size $N_B$

    **for** $k = 1, ..., K$ **do**
        Random sample batch data $\mathcal{D}_{\text{batch}} \subset \mathcal{D}$
        $\hat{\mathcal{C}}_{S^+|S}, \hat{\mathcal{C}}_{S|OH}$ = Algorithm 2 by using batch data $\mathcal{D}_{\text{batch}}$ and deep features
        Compute loss $l(\theta_S) = \|\hat{\mathcal{C}}_{S^+|S}\phi_{\theta_S}(s) - \phi_{\theta_S}(s^+)\|^2$
        Compute loss $l(\theta_O, \theta_H) = \|\hat{\mathcal{C}}_{S|OH}[\phi_{\theta_O}(o) \otimes \phi_{\theta_H}(h)] - \phi_{\theta_S}(s)\|^2$
        Compute loss $l(\theta_S, \theta_S') = \|\phi_{\theta_S'}^\dagger(\phi_{\theta_S}(s)) - s\|^2$
        Update the deep features.
        $\theta_S = \theta_S + \alpha\nabla_{\theta_S}l(\theta_S)$;
        $\theta_O, \theta_H = \theta_O + \alpha\nabla_{\theta_O}l(\theta_O, \theta_H), \theta_H + \alpha\nabla_{\theta_H}l(\theta_O, \theta_H)$;
        $\theta_S, \theta_S' = \theta_S + \alpha\nabla_{\theta_S}l(\theta_S, \theta_S'), \theta_S' + \alpha\nabla_{\theta_S'}l(\theta_S, \theta_S')$
    **end for**
    Compute $\hat{\mathcal{C}}_{S^+|S}, \hat{\mathcal{C}}_{S|OH}$ = Algorithm 2 by using the whole dataset $\mathcal{D}$ and trained deep features $\phi_{\theta_S}, \phi_{\theta_O}, \phi_{\theta_H}$.
    Compute the error covariance matrices $\mathcal{B}, \mathcal{R}, \mathcal{Q}$ from Subsection D.1
    **Return** $\phi_{\theta_S}, \phi_{\theta_O}, \phi_{\theta_H}, \hat{\mathcal{C}}_{S^+|S}, \hat{\mathcal{C}}_{S|OH}, \mathcal{B}, \mathcal{R}$, and $\mathcal{Q}$

---

**Algorithm 2** Tensor-Var training with kernel feature

---

**Require:** Dataset $\mathcal{D} = \{s_i, o_i, h_i, s_i^+\}_{i=1}^N$; kernel features $\phi_S(s) = k_S(s, \cdot), \phi_O(o) = k_O(o, \cdot), \phi_H(h) = k_H(h, \cdot)$
    Compute the Gram matrix $K_S$ where $[K_S]_{ij} = k_S(s_i, s_j)$
    Compute the Gram matrix $K_{OH}$ where $[K_{OH}]_{ij} = k_{OH}(o_i \otimes h_i, o_j \otimes h_j) = k_O(o_i, o_j)k_H(h_i, h_j)$
    If $N$ is too large, say $N \geq 10000$, using the Nystrom approximation to select a subset $\mathcal{D}^s = \{s_i, o_i, h_i, s_i^+\}_{i=1}^n$
    Compute the feature matrix $\Phi_S = [\phi_S(s_1), ..., \phi_S(s_n)]$
    Compute the feature matrix $\Phi_{S^+} = [\phi_S(s_1^+), ..., \phi_S(s_n^+)]$
    Compute the feature matrix $\Phi_{OH} = [\phi_{O,H}(o_1, h_1), ..., \phi_{O,H}(o_n, h_n)]$
    CME for the system dynamics. $\hat{\mathcal{C}}_{S^+|S} = \Phi_{S^+}(K_S + \lambda I)^{-1}\Phi_S$
    CME for the inverse observation model. $\hat{\mathcal{C}}_{S|OH} = \Phi_S(K_{OH} + \lambda I)^{-1}\Phi_{OH}^T$
    Compute the error covariance matrices $\mathcal{B}, \mathcal{R}, \mathcal{Q}$ from Subsection D.1.
    Fit the projection matrix for pre-image. $\hat{\mathcal{C}}_{\text{proj}} = \mathbf{S}(K_S + \lambda I)^{-1}\Phi_S^T$ where $\mathbf{S} = (s_1, ..., s_n)$
    **Return** $\hat{\mathcal{C}}_{S^+|S}, \hat{\mathcal{C}}_{S|OH}, \hat{\mathcal{C}}_{\text{proj}}, \mathcal{B}, \mathcal{R}$, and $\mathcal{Q}$

---

**Algorithm 3** Tensor-Var assimilation-forecasting

---

**Require:** assimilation window $\{o_t, h_t\}_{t=0}^T$; background state $s_b$; leading time $\tau$; kernel features $\phi_S, \phi_O, \phi_H$ (or trained deep features $\phi_{\theta_S}, \phi_{\theta_O}, \phi_{\theta_H}$); CME operators $\hat{\mathcal{C}}_{S^+|S}$ and $\hat{\mathcal{C}}_{S|OH}$; Error covariance matrices $\mathcal{B}, \mathcal{R}, \mathcal{Q}$.
    Perform Quadratic Programming with objective

$$\min_{\{z_t\}_{t=0}^T} \|z_0 - \phi_S(s_0^b)\|_{\mathcal{B}^{-1}}^2 + \sum_{t=0}^T \|z_t - \hat{\mathcal{C}}_{S|OH}\phi_{OH}(o_t, h_t)\|_{\mathcal{R}^{-1}}^2$$
$$+ \sum_{t=0}^{T-1} \|z_{t+1} - \hat{\mathcal{C}}_{S^+|S}z_t\|_{\mathcal{Q}^{-1}}^2,$$

    Project back to original space with $\hat{s}_t = \hat{\mathcal{C}}_{\text{proj}}z_t$ (or using learned inverse feature $\hat{s}_t = \phi_{\theta_S'}^\dagger(z_t)$)
    **for** $t = 1, ..., \tau$ **do**
        Predict $z_{t+T} = \hat{\mathcal{C}}_{S^+|S}z_{T+t-1}$
        Project back to original space with $\hat{s}_t = \hat{\mathcal{C}}_{\text{proj}}z_t$ (or inverse feature $\hat{s}_{T+t} = \phi_{\theta_S'}^\dagger(z_{T+t})$)
    **end for**

---

# E. Experiment Settings

**Training details.** Given the generated data, we construct two datasets: $\mathcal{D}_{\text{dyn}} = \{\{(s_t^i, s_{t+1}^i)\}_{t=0}^{T-1}\}_{i=1}^{N}$ and $\mathcal{D}_{\text{obs}} = \{\{(s_t^i, o_t^i, h_t^i)\}_{t=0}^{T}\}_{i=1}^{N}$. The DFs are trained in two steps using these datasets. First, the state DFs $\phi_{\theta_S}, \phi_{\theta_S'}^\dagger$ are trained on $\mathcal{D}_{\text{dyn}}$ and we store the estimated operator $\hat{\mathcal{C}}_{S^+|S}$. Next, with the state features fixed, the observation DF $\phi_{\theta_O}$ and history DF $\phi_{\theta_H}$ are trained on $\mathcal{D}_{\text{obs}}$ according to (3.2), storing the estimated operator $\hat{\mathcal{C}}_{S|OH}$. The baseline method (Frerix et al., 2021) is trained on $\mathcal{D}_{\text{obs}}$, excluding history. All models are trained with the Adam optimizer (Kingma, 2014) for 200 epochs, using batch sizes from 256 to 1024 for stable operator estimation. Additional details on the DFs, baselines, and training procedures can be found in Appendix E.

**Implementation details.** For all baseline methods, we employ the L-BFGS algorithm for Variational Data Assimilation (Var-DA) optimization, implemented in JAX (Bradbury et al., 2018). The 4D-Var baselines use numerical dynamical models based on the 8th-order Runge-Kutta method and the 4th-order Exponential Time Differencing Runge-Kutta (ETDRK) method (Cox & Matthews, 2002) for the Lorenz-96 and KS systems. For Tensor-Var, we apply interior-point quadratic programming to solve the linearized 4D-Var optimization, using CVXPY (Diamond & Boyd, 2016). All training is conducted on a workstation with a 48-core AMD 7980X CPU and an Nvidia GeForce 4090 GPU.

## E.1. Lorenz 96

First, we consider the single-level Lorenz-96 system, which is introduced in (Lorenz, 1996) as a low-order model of atmospheric circulation along a latitude circle. The system state is $[S_1, ..., S_K]$ representing atmospheric velocity at $K$ evenly spaced locations and is evolved according to the governing equation:

$$\frac{dS_k}{dt} = -S_{k-1}(S_{k-2} - S_{k+1}) - S_k + F,$$

with periodic boundary conditions $x_{k+K} = x_k$. The first term models advection, and the second term represents a linear damping with magnitude $F$. In general, the dynamics become more turbulent/chaotic as F increases. We choose the number of variables $K = 40, 80$ and the external forcing $F = 10$, where the system is chaotic with a Lyapunov time of approximately 0.6 time units. As an observation model for the following experiments, we randomly observe $25\%$ states (e.g. 10 in $K = 40$). Our models are trained on a dataset $\mathcal{D}$ of $N = 100$ trajectories, each trajectory consisting of $= 5000$ time steps, generated by integrating the system from randomly sampled initial conditions.

### E.1.1. DATA GENERATION.

To generate the dataset, we use the 8th-order Runge-Kutta (Butcher, 1996) method to numerically integrate the Lorenz-96 systems with sample step 0.1 and the integration step $\Delta t$ size is set to 0.01. The system is integrated from randomly sampled initial conditions, and data is collected once the system reaches a stationary distribution. For an observation operator, we use subsampling which every 5th and 10th variable for 40 and 80-dimensional system are observed via the nonlinear mapping $5 \arctan(\cdot \pi/10) + \epsilon$ with noise $\epsilon \sim \mathcal{N}(0, 0.1)$ (see Figure 6 in Appendix E.1 for an example). The arctan: $\mathbb{R} \mapsto [-\frac{\pi}{2}, \frac{\pi}{2}]$ squeezes the state variable $S_k$ into $[-\frac{\pi}{2}, \frac{\pi}{2}]$, which is difficult for inverse estimation. We integrate the Lorenz96 system with observation interval $\Delta t = 0.1$. The history length is set as 10 such that $h_t = (o_{t-10}, ..., o_{t-1})$.

### E.1.2. ADDITIONAL EXPERIMENT RESULTS.

We provide qualitative results in Figure 7 for the Lorenz 96 system at two different dimensions: 40 (left) and 80 (right). Each subplot illustrates the normalized absolute error for various methods, including 3D-VAR, 4D-VAR, (Frerix et al., 2021), and Tensor-Var, compared to the ground truth. The assimilation window length is set to 5 (indicated by the red dashed line), with forecasts extended for an additional 100 steps based on the assimilated results. Tensor-Var generally outperforms the other methods in both dimensions and maintains stable long-term forecasts, comparable to other model-based approaches. For 3D- and 4D-VAR with partially observed models, the observed states show minimized errors (indicated by the dark lines), while the errors of unobserved states remain uncontrollable, as clearly shown in Figure 7.

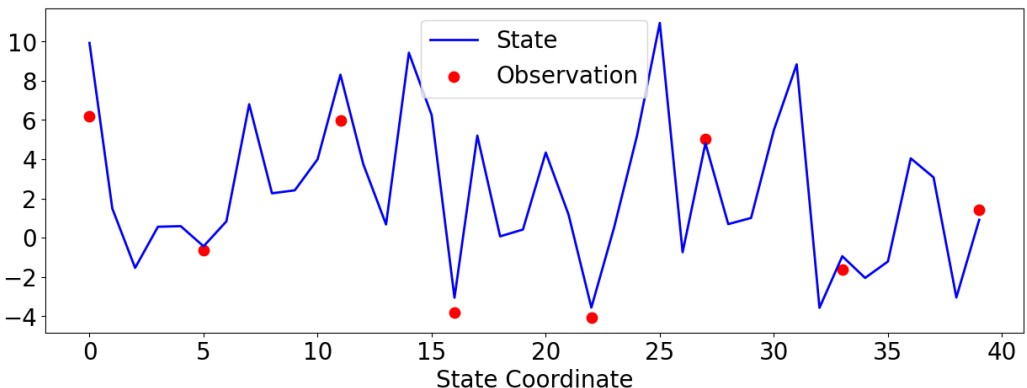

*Figure 6.* Nonlinear observation model in the Lorenz 96 system: the state values are represented by the solid blue curve, with the observed grid points indicated by red dots.

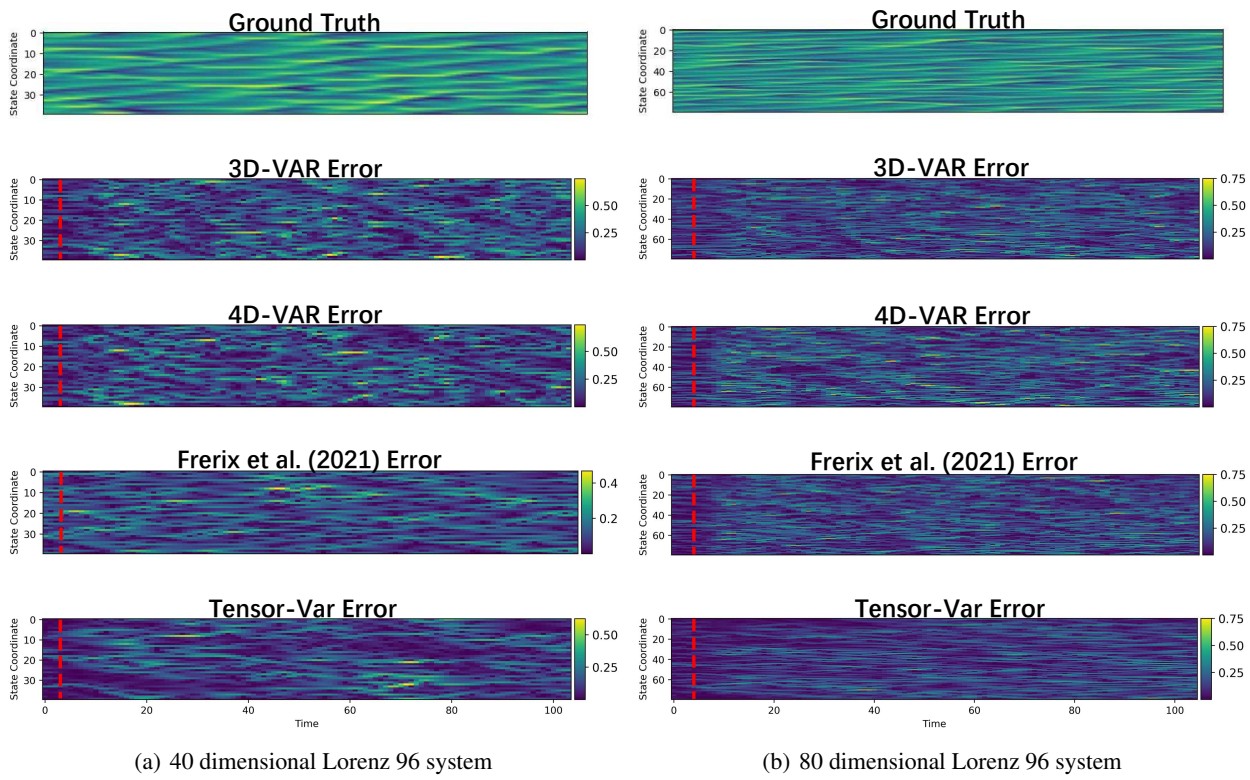

(a) 40 dimensional Lorenz 96 system             (b) 80 dimensional Lorenz 96 system

*Figure 7.* Qualitative error comparison for the Lorenz 96 system at (a) 40 dimensions and (b) 80 dimensions. The plots show the normalized absolute errors for various methods, including 3D-VAR, 4D-VAR, Frerix et al. (2021), and Tensor-Var, compared to the ground truth. The assimilation window length is set to 5 (indicated by the red dashed line), with forecasts extended for an additional 100 steps based on the assimilated results.

### E.2. Kuramoto-Sivashinsky

Next, we consider the Kuramoto-Sivashinsky (KS) equation, a nonlinear PDE system known for its chaotic behaviour and widely used to study instability in fluid dynamics and plasma physics (Papageorgiou & Smyrlis, 1991). The dynamics in spatial domain $u(x, t)$ is given by,

$$\frac{\partial u}{\partial t} + u \frac{\partial u}{\partial x} + \frac{\partial^2 u}{\partial x^2} + \frac{\partial^4 u}{\partial x^4} = 0,$$

where $x \in [0, L]$ with periodic boundary conditions. We set the domain length $L = 32\pi$, large enough to induce complex patterns and temporal chaos due to high-order term interactions (Cvitanović et al., 2010). The system state $u(x, t)$ was discretized into $n_s = 128$ and $n_s = 256$. The observation model is the same as in Lorenz-96, where $25\%$ states can be observed. In this case, our models are trained on a dataset $\mathcal{D}$ consisting of $N = 100$ trajectories, each with $L = 5000$ time steps and a discretization of $\Delta t = 0.01$, sampled from the stationary distribution with different initial conditions.

#### E.2.1. DATA GENERATION.

To generate the dataset, we use the exponential time differencing Runge–Kutta method (ETDRK), which has proven effective in computing nonlinear partial differential equation (Cox & Matthews, 2002) with an integration step $\Delta t = 0.001$ and sample step $0.01$. The system is integrated from randomly sampled initial conditions, and data is collected once it reaches a stationary distribution. For observations, we use subsampling, observing every 8th state in both 128- and 256-dimensional systems, we use subsampling which every 8th for both 128 and 256-dimensional system are observed with noise $\epsilon \sim \mathcal{N}(0, 1)$, and $5 \arctan(\cdot \pi / 10)$ as nonlinear mapping (see Figure 6 in Appendix E.1 for an example). The history length is set to 10 as well.

#### E.2.2. ADDITIONAL EXPERIMENT RESULTS.

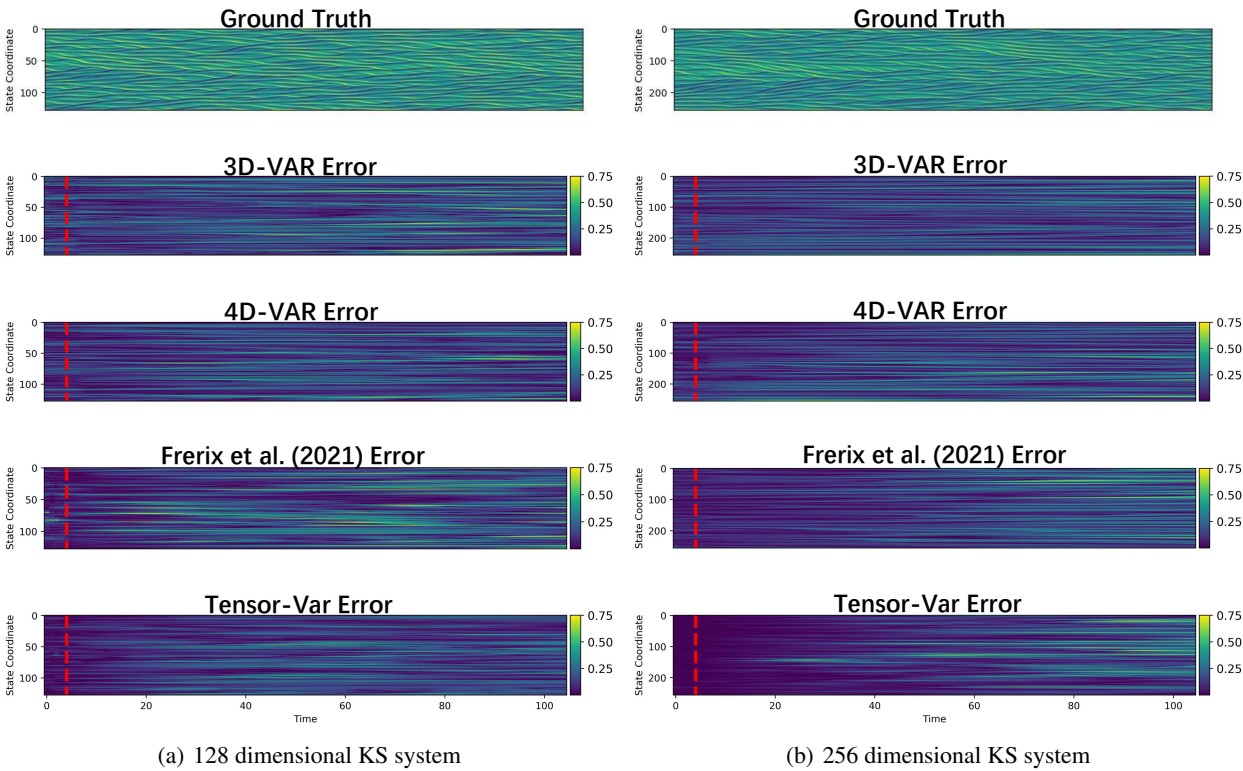

(a) 128 dimensional KS system    (b) 256 dimensional KS system

*Figure 8.* Qualitative error comparison for the KS system at (a) 128 dimensions and (b) 256 dimensions. The plots show the normalized absolute errors for various methods, including 3D-VAR, 4D-VAR, Frerix et al. (2021), and Tensor-Var, compared to the ground truth. The assimilation window length is set to 5 (indicated by the red dashed line), with forecasts extended for an additional 100 steps based on the assimilated results.

We provide qualitative results in Figure 8 for the KS systems at two different dimensions: 128 (left) and 256 (right). Each subplot illustrates the normalized absolute error for various methods, including 3D-VAR, 4D-VAR, Frerix et al. (2021), and Tensor-Var, compared to the ground truth. The assimilation window length is set to 5 (indicated by the red dashed line), with forecasts extended for an additional 100 steps based on the assimilated results.

Compared to the Lorenz-96 system, the KS system is more complex, being governed by partial differential equations (PDEs) that account for spatial evolution. In both dimensions, Tensor-Var consistently outperforms other methods, particularly in capturing chaotic dynamics during the initial forecast phase. It also maintains long-term stability in a more complex PDE system, comparable to other model-based approaches. In contrast, 3D-VAR struggles with assimilation, especially in the 256-dimensional case, due to its inability to capture temporal evolution, leading to rapid error divergence. This underscores the critical importance of temporal modeling in chaotic systems. A similar pattern of error between observed and unobserved states is evident in Figure 8.

### E.3. Global NWP

We consider a global numerical weather prediction (NWP) problem using the European Centre for Medium-Range Weather Forecasts (ECMWF) Atmospheric Reanalysis (ERA5) dataset (Hersbach et al., 2020). This dataset provides high-resolution atmospheric reanalysis data from 1940 to the present, offering the most comprehensive estimate of atmospheric dynamics. For our proof of concept, we focus on five upper level physical variables: 500 hPa geopotential height, 850 hPa temperature, 700 hPa humidity, and 850 hPa wind speed (meridional and zonal components) at a spatial resolution of $64 \times 32$.

The data is sourced from the WeatherBench2 repository (Rasp et al., 2024). From this dataset, we randomly sample grid points with 15% spatial coverage. The sampled observations include additive noise equivalent to 0.01 times the standard deviation of the state variable, ensuring robustness against observational uncertainty (see Figure 9). For model training, we use ERA5 data from 1979-01-01 to 2016-01-01, separating data from post-2018 for testing. There are 51,100 consecutive system states with generated observations for training and 2,920 data for testing.

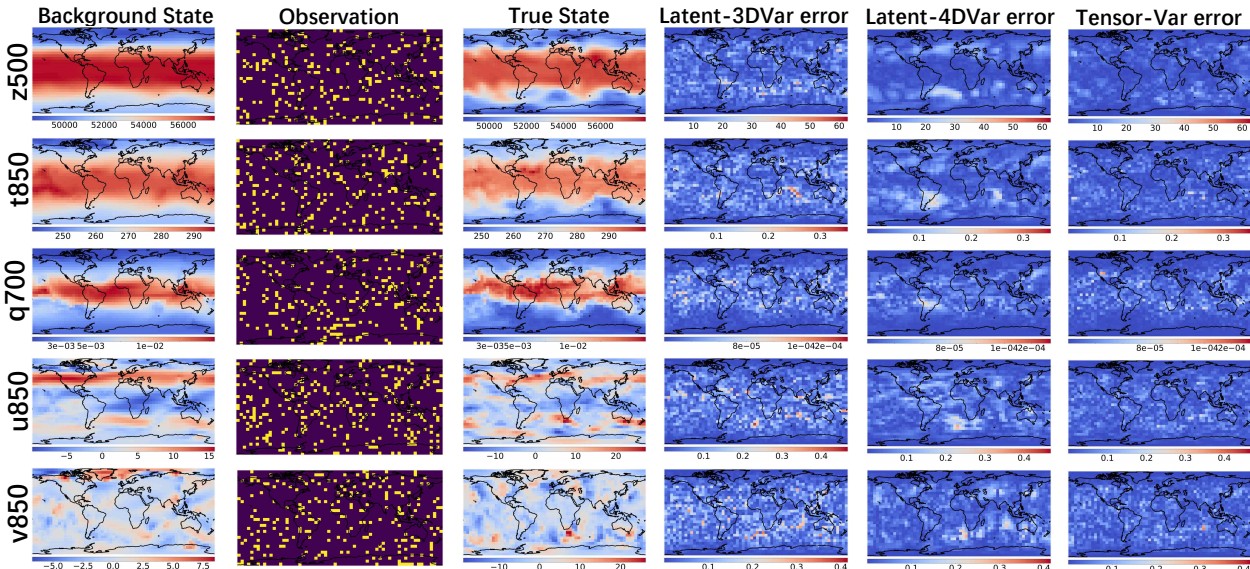

*Figure 9.* Visualization of assimilation results for five variables from ERA5 data at time 2018-01-01 00:00. Each column (from left to right) displays the background state, observations, true state, and errors for Latent-3DVar, Latent-4DVar, and Tensor-Var. The reported error was a weighted absolute error in each pixel; see Appendix E.3 for more details.

In addition to the results presented in the main experiments, we evaluate the forecasting quality of Tensor-Var based on the assimilated state. Figure 10 shows the mean RMSE weighted by latitude (Rasp et al., 2024) for five variables predicted by Tensor-Var at various lead times $\tau$, where $\tau = 0$ represents the assimilation error in the final state of the assimilation window.

*Figure 10.* The (non-cyclic) forecasting quality of Tensor-Var in NWP experiments with leading time zero as the final state in the assimilation window, is evaluated across different experiments. The five sub-figures display the NWP forecast for 5 variables (15-day in total).

**Area-weighting Root mean squared error (RMSE).** The error is defined for each variable and level as

$$\sqrt{\frac{1}{TIJ}\sum_{t=1}^{T}\sum_{i=1}^{I}\sum_{j=1}^{J}(w(i)\hat{s}_{t,i,j} - s_{t,i,j})^2},$$

which is area-weighting over grid points. This is because on an equiangular latitude-longitude grid, grid cells at the poles have a much smaller area compared to grid cells at the equator. Weighing all cells equally would result in an inordinate bias towards the polar regions. The latitude weights $w(i)$ are computed as:

$$w(i) = \frac{\sin(\theta_i^u) - \sin(\theta_i^l)}{\frac{1}{I}\sum_{i=1}^{I}(\sin(\theta_i^u) - \sin(\theta_i^l))},$$

$\theta_u^i$ and $\theta_l^i$ indicate upper and lower latitude bounds, respectively, for the grid cell with latitude index $i$.

## E.4. ASSIMILATION FROM SATELLITE OBSERVATIONS

### E.4.1. DATA GENERATION.

We collect the weather satellite track data from https://celestrak.org/NORAD/elements/ for the period 1979-01-01 00:00:00 to 2020-01-01 00:00:00. Observations are matched to the high-resolution ERA5 dataset ($240 \times 121$ grid) by identifying the nearest neighborhood grid points along the satellite track to generate observations. Furthermore, we use an observation frequency of up to every half hour within the 2 hours before each assimilation time and add white noise with a standard deviation of 1% of the standard deviation of the corresponding state variables.

### E.4.2. ANOMALY CORRELATION COEFFICIENT EVALUATION

In addition to the NRMSE results reported in Subsection 4.3, we also evaluate the assimilated fields using the Anomaly Correlation Coefficient (ACC), which measures the pattern similarity between anomalies in the analysis (or forecast) and a reference field relative to a climatological mean. This metric is widely used in numerical weather prediction to assess the accuracy of large-scale spatial patterns. We compare the ACC performance of our method with the SOTA data-driven baseline, FengWu 4D-Var (Xiao et al., 2024). ACC is formally defined as:

$$\text{ACC} = \frac{\sum_t (\hat{s}_t - \bar{s})(s_t - \bar{s})}{\sqrt{\sum_t (\hat{s}_t - \bar{s})^2}\sqrt{\sum_t (s_t - \bar{s})^2}},$$

where $\hat{s}$ is the result of assimilation, $s$ is the corresponding reference value (e.g. ERA5), and $\bar{s}$ is the climatological mean. Figure 11 reports the mean and standard deviation of ACC for five atmospheric variables over a 7-day forecast horizon, with an assimilation window of length 5. The results indicate that Tensor-Var consistently achieves higher ACC values than FengWu 4D-Var over a long horizon.

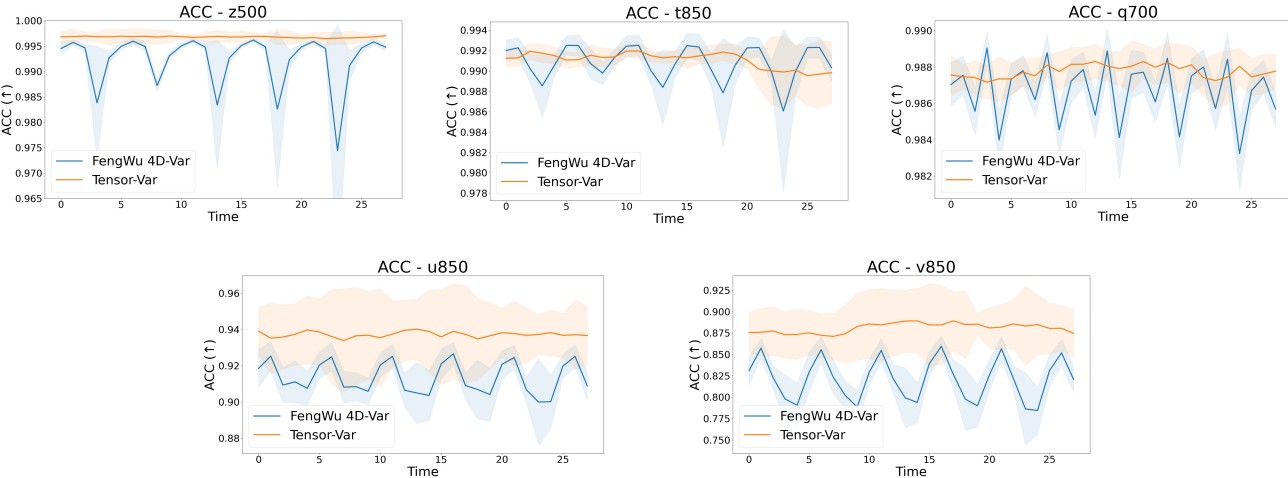

*Figure 11.* Comparison of ACC over a 7-day horizon for five atmospheric variables from Fengwu 4D-Var and Tensor-Var. Each time-step represents a 6-hour interval.

### E.4.3. LONG ROLL-OUT STABILITY TEST

To assess the long-term stability of our method, we conduct a one-year roll-out experiment under the following settings: spatial resolution of $240 \times 121$; meteorological variables {z500, t850, q700, u850, v850}; a 6-hour time step; assimilation window length of 5 steps; and a forecast lead time of 7 days. The experiment covers the full year from January 1 to December 31, 2018, with training and evaluation settings consistent with Section 4.3.

Figure 12 illustrates the long-term behavior of Tensor-Var in comparison to FengWu 4D-Var. The results show that Tensor-Var effectively controls the estimation error over extended periods and achieves lower error magnitudes than FengWu 4D-Var throughout most of the year. However, we observe that Tensor-Var exhibits instability after approximately 800 assimilation steps, whereas FengWu maintains stable performance.

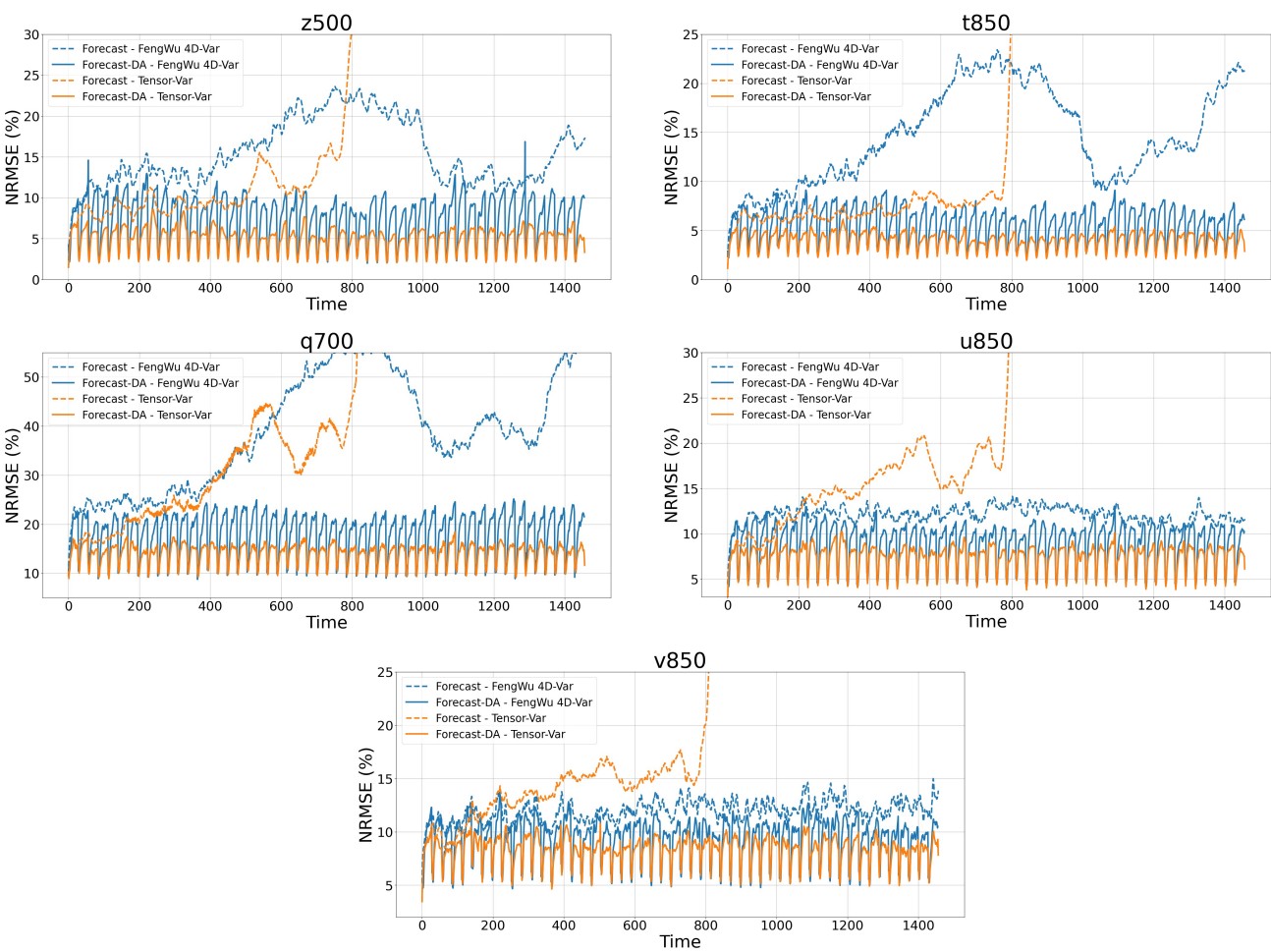

*Figure 12.* Long-term roll-out stability test for the five meteorological variables over a one-year horizon. The plot compares the NRMSE of forecasts (dashed) and forecast-DA (solid) outputs from Tensor-Var and FengWu 4D-Var.

The observed instability can be attributed to the linear dynamical structure of Tensor-Var, which is able to handle short-term forecasting but can be insufficient for capturing nonlinear long-term dynamics. In contrast, the transformer-based architecture of FengWu 4D-Var appears to be more robust for extended temporal integration. These findings highlight a trade-off between assimilation efficiency and long-horizon stability. Future work will focus on improving Tensor-Var's dynamical expressiveness while preserving its computational advantages.

### E.4.4. ADDITIONAL QUALITATIVE RESULTS

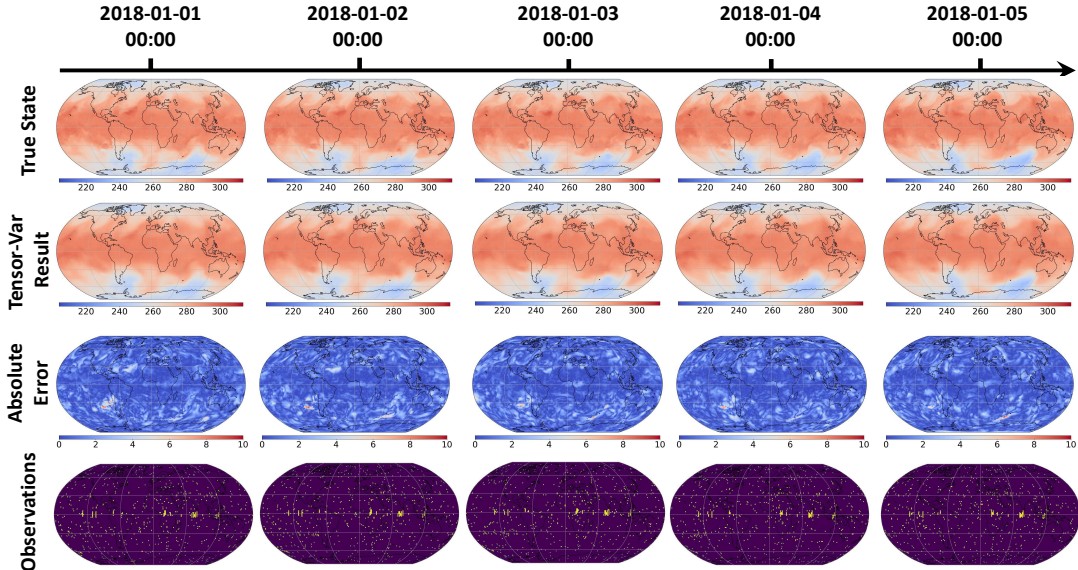

*Figure 13.* Visualization of continuous assimilation results, absolute errors, and observation locations for t850 (temperature), starting from 2018-01-01 00:00.

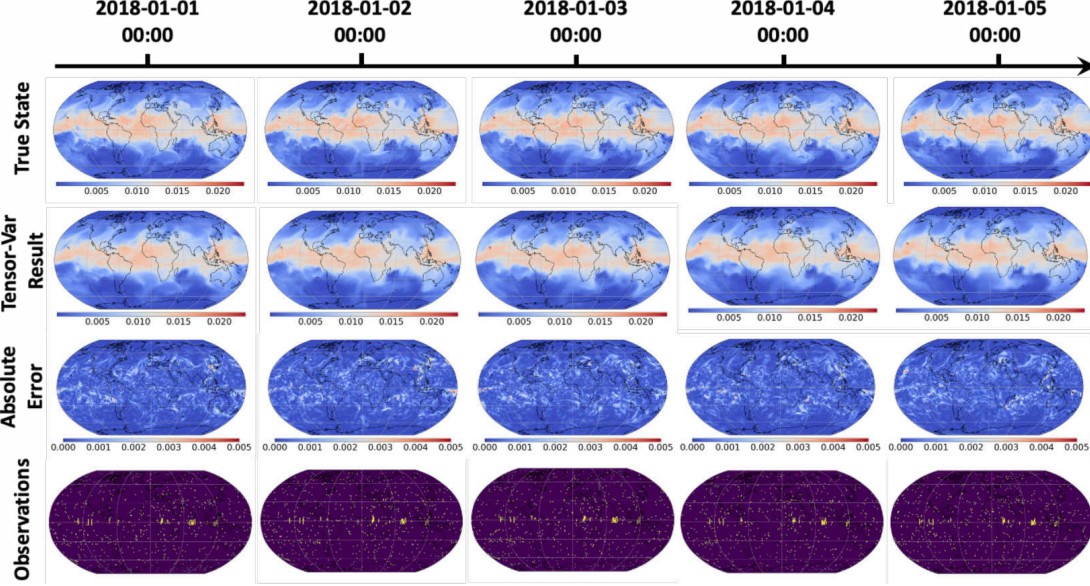

*Figure 14.* Visualization of continuous assimilation results, absolute errors, and observation locations for q700 (humidity), starting from 2018-01-01 00:00.

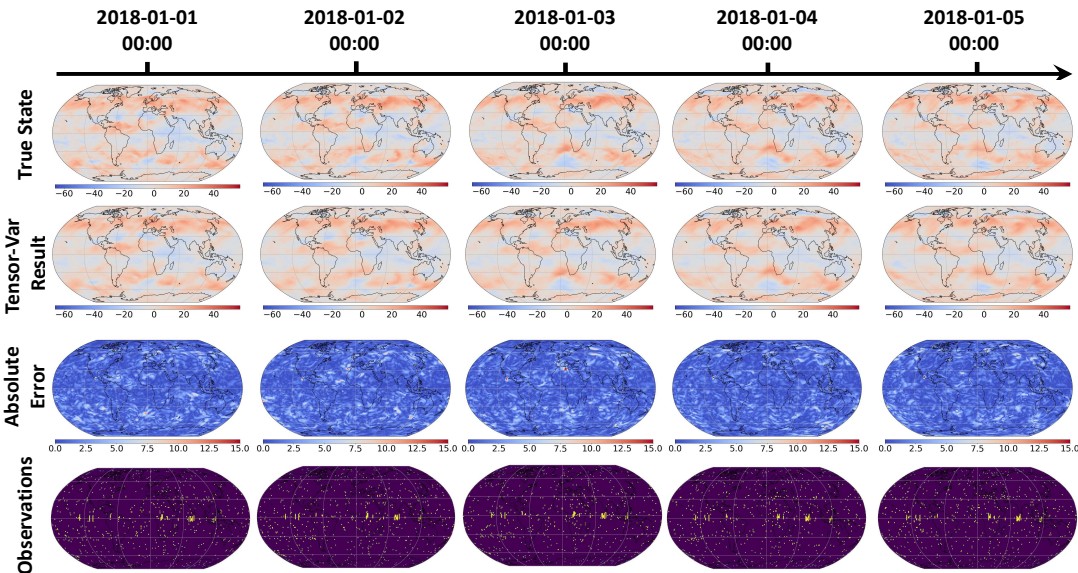

*Figure 15.* Visualization of continuous assimilation results, absolute errors, and observation locations for u850 (meridional wind speed), starting from 2018-01-01 00:00.

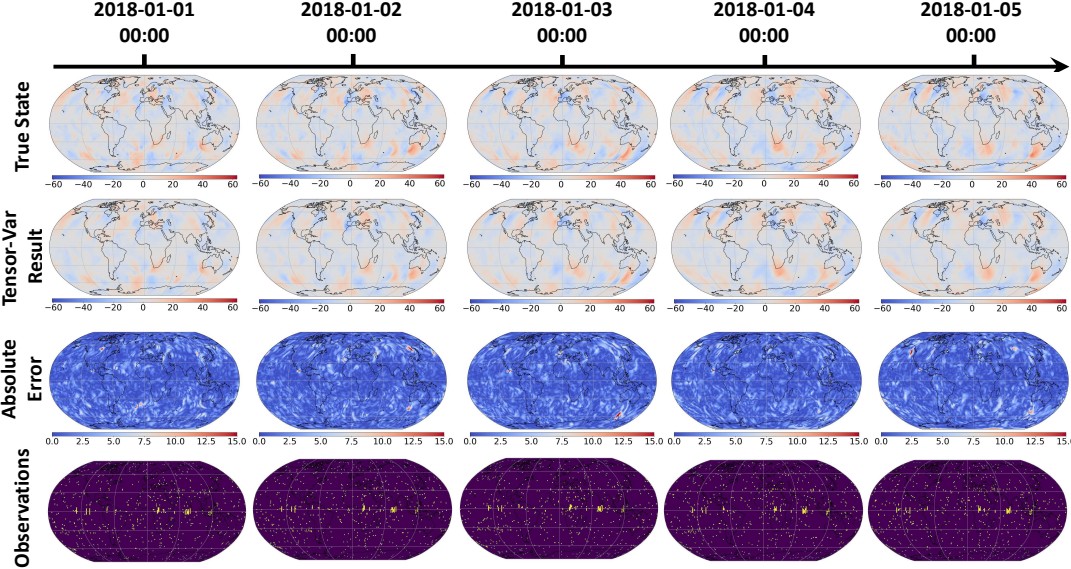

*Figure 16.* Visualization of continuous assimilation results, absolute errors, and observation locations for v850 (zonal wind speed), starting from 2018-01-01 00:00.

## E.5. Ablation study

In this section, we provide the details of the ablation studies of (1) the history length $m$, (2) the dimensions of the feature dimension and comparison with standard kernel functions, and (3) the effects of the estimated error matrices. All ablation studies are conducted on Lorenz-96 systems with $n_s = 40$ and $n_s = 80$. We fix the remaining hyperparameters consistent with the main experiments and vary only the parameters under investigation.

**(1) Effect of the history length $m$.** We examine the impact of the history length $m$ on learning the inverse observation operator and its effect on state estimation accuracy. The feature dimensions $d_s$, $d_o$, and $d_h$ remain constant, while the history length is varied by adjusting the size of the final linear layer (see Table 5). The state feature dimensions $d_s$ are set to 60 and 120 for the two system dimensions.

**(2) Effect of DFs.** We implement Tensor-Var with a Gaussian kernel using kernel PCA projected to the first 60 and 120 eigen-coordinates in scikit-learn (Pedregosa et al., 2011) by aligning with the dimension of the used DFs. For the Gaussian kernel, we approach the pre-image problem by fitting a projection operator (Kwok & Tsang, 2004). The space $\mathcal{S} \subset \mathbb{R}^{n_s}$ together with the linear kernel $k(s_i, s_j) = s_i^T s_j$ also forms an RKHS. Therefore, we can define the projection operator as a CME operator that maps from the feature space $\mathbb{H}_S$ to the original space $\mathcal{S}$ as $\hat{\mathcal{C}}_{\text{proj}} = \mathbf{S}(K_S + \lambda I)^{-1}\Phi_S^T$, where $\mathbf{S} = [s_1, ..., s_n]$. By applying $\hat{\mathcal{C}}_{\text{proj}}$ now to the state, we can obtain the mean estimation of the kernel mean embedding in $\mathcal{S}$ such that $\hat{s} = \hat{\mathcal{C}}_{\text{proj}}\mu_{\mathbb{P}_S} = \mathbb{E}_S[\hat{\mathcal{C}}_{\text{proj}}\phi_S(S)] = \mathbb{E}_S[S]$. We normalize the dataset to a standard Gaussian distribution and select the length scale $\gamma = 1.0$ by performing a cross-validation on $\gamma = [0.5, 0.75, 1.0, 1.25, 1.5, 2]$.

## E.6. Model architecture

*Table 5.* Deep feature architecture for 1D chaotic systems with dimensions $n_s, n_o, n_h = m \times n_o$ and feature dimension $d_s, d_o, d_h$

| Components | Layer | Weight size | Bias size | Activation |
|---|---|---|---|---|
| $\phi_{\theta_S}$ | Fully Connected | $n_s \times 4n_s$ | $4n_s$ | Tanh |
| | Fully Connected | $4n_s \times 2n_s$ | $2n_s$ | Tanh |
| | Fully Connected | $2n_s \times d_s$ | $d_s$ | |
| $\phi_{\theta_S'}^{\dagger}$ | Fully Connected | $d_s \times 2n_s$ | $2n_s$ | Tanh |
| | Fully Connected | $2n_s \times 4n_s$ | $4n_s$ | Tanh |
| | Fully Connected | $4n_s \times n_s$ | $n_s$ | |
| $\phi_{\theta_O}$ | Fully Connected | $n_o \times 4n_o$ | $4n_o$ | Tanh |
| | Fully Connected | $4n_o \times 2n_o$ | $2n_o$ | Tanh |
| | Fully Connected | $2n_o \times d_o$ | $d_o$ | |
| $\phi_{\theta_H}$ | Convolution 1D | $m \times 2m \times 5$ | $2m$ | Tanh |
| | Max Pooling (size=2) | | | |
| | Convolution 1D | $2m \times 4m \times 3$ | $4m$ | Tanh |
| | Max Pooling (size=2) | | | |
| | Flatten | | | |
| | Fully Connected | $mn_o \times d_h$ | $d_h$ | |

*Table 6.* Model architecture for Global NWP with input dimension $(H, W, C)$ with $C$ physical variables and spatial resolution $H \times W$. The implementation of vision Transformer (ViT) block follows (Zamir et al., 2022) with applications in DA followed by (Nguyen & Fablet, 2024).

| Components | Layer | Layer number | $C, (H, W)$ | Activation |
|---|---|---|---|---|
| $\phi_{\theta_S}$ | Convolution2d | 1 | $C \to 4C, (H, W)$ | |
| | Transformer Block | 2 | $4C \to 4C, (\frac{H}{2}, \frac{W}{2})$ | ReLU |
| | Transformer Block | 3 | $4C \to 8C, (\frac{H}{4}, \frac{W}{4})$ | ReLU |
| | Transformer Block | 3 | $8C \to 8C, (\frac{H}{8}, \frac{W}{8})$ | ReLU |
| | Flatten | | $(8C, \frac{H}{8}, \frac{W}{8}) \to \frac{CHW}{8}$ | |
| | Fully Connected | 1 | $\frac{CHW}{8} \to d_s$ | |
| $\phi_{\theta'_S}^{\dagger}$ | Fully Connected | 1 | $d_s \to \frac{CHW}{8}$ | |
| | Transpose | | $\frac{CHW}{8} \to (8C, \frac{H}{8}, \frac{W}{8})$ | |
| | Transformer Block | 3 | $8C \to 8C, (\frac{H}{8}, \frac{W}{8})$ | ReLU |
| | Transformer Block | 3 | $8C \to 4C, (\frac{H}{4}, \frac{W}{4})$ | ReLU |
| | Transformer Block | 2 | $4C \to 4C, (\frac{H}{2}, \frac{W}{2})$ | ReLU |
| | Convolution2d | 1 | $4C \to C, (H, W)$ | |
| $\phi_{\theta_O}$ | Convolution2d | 1 | $C \to 2C, (H, W)$ | |
| | Transformer Block | 2 | $2C \to 2C, (\frac{H}{2}, \frac{W}{2})$ | ReLU |
| | Transformer Block | 3 | $2C \to 4C, (\frac{H}{8}, \frac{W}{8})$ | ReLU |
| | Flatten | | $(4C, \frac{H}{8}, \frac{W}{8}) \to \frac{CHW}{16}$ | |
| | Fully Connected | 1 | $\frac{CHW}{16} \to d_o$ | |
| $\phi_{\theta_O}$ | Convolution2d | 1 | $mC \to 2C, (H, W)$ | |
| | Transformer Block | 2 | $2C \to 2C, (\frac{H}{2}, \frac{W}{2})$ | ReLU |
| | Transformer Block | 3 | $2C \to 4C, (\frac{H}{8}, \frac{W}{8})$ | ReLU |
| | Flatten | | $(4C, \frac{H}{8}, \frac{W}{8}) \to \frac{CHW}{16}$ | |
| | Fully Connected | 1 | $\frac{CHW}{16} \to d_h$ | |

