# OpenReview forum: "Tensor-Var: Efficient Four-Dimensional Variational Data Assimilation"
_ICML.cc/2025/Conference — ICML 2025 poster_

### Official Review · Reviewer_wFvT · 2025-03-12

**Overall Recommendation:** 3

**Summary:**

The paper introduces Tensor-Var, a novel framework for variational data assimilation, which integrates kernel conditional mean embedding (CME) with four-dimensional variational assimilation (4D-Var). Traditional 4D-Var methods are computationally expensive and struggle with nonlinear dynamics and imperfect state-observation mappings. Tensor-Var addresses these challenges by linearizing nonlinear dynamics in a learned feature space, making the optimization convex while ensuring theoretical guarantees for consistent assimilation results.

To enhance scalability, the framework incorporates deep features (DFs) learned via neural networks, allowing it to handle large-scale problems efficiently. Experiments on chaotic systems and global weather prediction demonstrate that Tensor-Var significantly outperforms conventional and deep learning (DL)-based 4D-Var methods in accuracy while achieving a 10- to 20-fold speedup.

**Claims And Evidence:**

Yes.

**Essential References Not Discussed:**

NA

**Experimental Designs Or Analyses:**

Yes.

**Methods And Evaluation Criteria:**

Yes.

**Other Comments Or Suggestions:**

NA

**Other Strengths And Weaknesses:**

NA

**Questions For Authors:**

1. How sensitive is the performance of Tensor-Var to the choice of the kernel function in the conditional mean embedding (CME)? Have you tested different kernels, and how do they impact generalization across different dynamical systems?
2. While Tensor-Var achieves a 10- to 20-fold speedup, how does its computational complexity scale with increasing data dimensions and observation frequencies? Are there potential bottlenecks in large-scale real-world applications?

**Relation To Broader Scientific Literature:**

NA

**Theoretical Claims:**

No.

---

> ### Author Rebuttal · Authors · 2025-04-01
>
> We sincerely thank reviewer wFvT for these insightful questions. Below, we address each point in detail.
>
> - **Choice of kernel functions**
>
>    Thank you for raising the question about choices of kernel functions in Tensor-Var.  We would like to clarify that we have included an ablation study in Section 4.4 Table 3  on the Lorenz96 systems with $n_s=40$ and $n_s=80$. To further address the reviewer's comments, we also tested other kernel features, including Matern-1.5 and Matern-2.5. The results are provided in the following Table:
>
>   **Table: Comparison of different features with NRMSE (%) as the metric for state estimation accuracy.**
>   | Features/Experiments  | Lorenz-96 $n_s=40$ | Lorenz-96 $n_s=80$|
>   |-----------------------|----------------------|----------------------|
>   | $d_s=20$            | 16.7±2.1              | 17.3±2.6              |
>   | $d_s=40$            | 14.1±1.3              | 16.8±2.2              |
>   | $d_s=60$            | 8.3±0.9               | 11.4±0.9              |
>   | $d_s=120$          | 9.7±0.7               | **9.0**±0.9              |
>   | Gaussian Kernel       | 8.4±0.5               | 14.3±1.4              |
>   | Matern-1.5            | 9.6±0.8               | 13.2±1.1              |
>   | Matern-2.5            | **7.2±0.2**           | 9.7±**0.7**           |
>
> In a low dimensional system ($n_s=40$), Matern-2.5 achieves the lowest mean and standard deviation. In the higher-dimensional setting ($n_s=80$), the deep feature with $d_s=120$ outperforms the other approaches, albeit with a slightly higher standard deviation compared to Matern-2.5. In general, the results suggest that fixed kernel features may be more reliable at lower dimensions, but deep features become advantageous as dimensionality increases.
>
> For high-dimensional weather forecasting problems, fixed kernel features tend to be poorly scaled and are inadequate for processing structured data. Thus, we propose deep features to address scalability challenges and extend Tensor-Var for broad applications.
>
>
> - **Computational Complexity and Scalability**
>
>      We thank the reviewer for their insightful question regarding the scalability of Tensor-Var. The computational complexity of Tensor-Var primarily depends on the feature dimension rather than the system dimension, as DA optimisation is performed within the feature space. As shown in Table 1 (Section 4.1), the evaluation time exhibits only a slight increase with the increasing system dimension.
>
>      Moreover, the experiments presented in Section 4.2 and Section 4.3 differ in spatial resolution, with Section 4.3 using a resolution of $240*121$, approximately 14 times higher than that of Section 4.2. Nonetheless, the computation time does not increase substantially.
>
>      One potential bottleneck arises from solving the least square problems to compute the two CME operators in high-dimensional feature spaces during training. Incorporating more efficient solvers, such as the Krylov subspace method [Liesen et al. (2013)], could address this challenge in future work.
>
> We hope these responses adequately address the reviewer’s concerns, and we look forward to any further comments you may have.
>
>
> **References**
>
>  - Liesen, Jörg, and Zdenek Strakos. Krylov subspace methods: principles and analysis. Numerical Mathematics and Scie, 2013.

---

### Official Review · Reviewer_wd4a · 2025-03-13

**Overall Recommendation:** 3

**Summary:**

This paper discusses Four-Dimensional Variational Data Assimilation (4D-Var), which is widely used in weather forecasting and dynamic system state estimation. Traditional methods struggle to properly model the nonlinear relationships between observational data and numerical models. The authors propose a novel approach that leverages Kernel Conditional Mean Embedding (CME) to linearize 4D-Var, transforming the nonlinear optimization problem into a convex optimization problem, thereby reducing the computational cost of optimization. Additionally, by incorporating past observational data, the proposed method enables robust estimation even in missing observations. Furthermore, the use of Deep Features (DFs) enhances scalability. Through experiments on two benchmark systems, as well as data assimilation experiments using global Numerical Weather Prediction (NWP) data and satellite observational data, the authors demonstrate that the proposed method achieves lower computational cost and reduced error compared to conventional 4D-Var methods.

## update after rebuttal

Thank you for the detailed response. I find the integration with AI-based forecasting models particularly interesting. I also appreciate the addition of the comparison method. Overall, I believe the contribution is significant, so I will keep my evaluation as it is. I mistakenly posted my comment in the Official Comment section. My apologies.

**Claims And Evidence:**

The authors propose, for the first time, a method that utilizes Kernel Conditional Mean Embedding (CME) to address the high computational cost and inability to properly handle nonlinear relationships in 4D-Var. Furthermore, by leveraging DFs, they achieve a practically feasible computational cost. This idea is highly logical and well-founded. Additionally, the effectiveness of the proposed method is properly demonstrated through extensive experimental data.

**Essential References Not Discussed:**

To the best of my knowledge, prior studies related to 4D-Var, including the well-known GraphCast, have been appropriately cited.

**Experimental Designs Or Analyses:**

As mentioned in Methods and Evaluation Criteria, appropriate comparison methods are used, and the datasets chosen for evaluation are also suitable.

**Methods And Evaluation Criteria:**

The proposed method is compared against conventional 3D-Var, 4D-Var, and ML-based 4D-Var (Frerix et al., 2021). These are appropriate baseline methods for demonstrating the advancements of the proposed approach. Additionally, the datasets used for evaluation include chaotic nonlinear systems, making them suitable for assessing the advantages of the proposed method.

**Other Comments Or Suggestions:**

I have no additional comments.

**Other Strengths And Weaknesses:**

As mentioned above, this paper has the potential to be widely applied to large-scale spatiotemporal data modeling and could lead to significant breakthroughs in this field.

Regarding weaknesses, while recent machine learning-based methods, such as GraphCast and GNN-based approaches, are cited, there is little direct comparison with machine learning-based methods. This may be due to the high computational cost of directly learning nonlinear relationships, making comparisons difficult. However, providing at least some level of comparison or an explanation of the challenges in conducting such comparisons could enhance the credibility of the study.

**Questions For Authors:**

Could you explain the lack of comparison with recent machine learning-based methods, such as GraphCast and GNN-based approaches?

**Relation To Broader Scientific Literature:**

The proposed method has broad applicability in modeling large-scale spatiotemporal data. In addition to the weather and satellite observation data evaluated in this paper, it can also be applied to various domains, such as spatiotemporal estimation of air pollution levels, probabilistic forecasting of earthquakes, time series prediction of stock prices and exchange rates, and estimation of biosignals (e.g., EEG and ECG). This approach has the potential to drive significant breakthroughs across a wide range of scientific fields.

**Theoretical Claims:**

The specific theoretical framework, including the estimation of error covariance matrices using kernel covariance matrices, is detailed in the Supplementary Material. A broad review of the content confirms that the key points are well covered.

---

> ### Author Rebuttal · Authors · 2025-03-31
>
> We greatly appreciate reviewer's constructive feedback and recognition of the strengths of our work. Below, we address each of the questions in detail.
>
> - **Compare with sota ML-based DA methods.**
>
>   We thank reviewer for raising the questions regarding the comparison with recent ML-based methods.
>
>   - Most existing ML-based methods, such as GraphCast [Lam et al. (2023)], primarily focus on weather forecasting rather than data assimilation. Therefore, direct comparisons between our DA framework and these forecasting methods are not straightforward.
>
>   - Nevertheless, our proposed DA framework can be seamlessly integrated with advanced AI-based weather forecasting models. By incorporating real-time observations, our approach consistently corrects the AI-based weather forecasting models, providing a more reliable starting point for subsequent forecasts. Furthermore, GNN-based approaches can be directly integrated into our proposed DA framework as kernel features, enabling the effective assimilation of spatially irregular data. This integration would further improve the model's ability to handle complex spatial information within the DA context.
>
>   - We recognise the importance of comparing our method with state-of-the-art ML-based DA approaches, as also highlighted by Reviewer dYtv. To address this concern, our revised manuscript has added the new results from a state-of-the-art machine learning-based data assimilation method, FengWu 4D-Var [Xiao et al. (2023)]. The results and comparison details are provided in our response to Reviewer dYtv under the question *Compare with sota ML-based DA methods & Baseline selections* and are also presented in the accessible [anonymous Google Slides](https://docs.google.com/presentation/d/1SWXKhJL_4mAKVQlY4HAxQTl_79C94m-2TwgtkYIvuNw/edit?usp=sharing). We believe this addition improves the credibility of our study and provides a more comprehensive evaluation.
>
> We hope this response adequately addresses the reviewer’s concerns. We sincerely appreciate the reviewer’s invested time and effort in carefully reviewing our work and providing valuable, constructive feedback.
>
> **References**
>   - Lam, Remi, et al. "Learning skillful medium-range global weather forecasting." Science 382.6677 (2023): 1416-1421.
>   - Xiao, Yi, et al. "Fengwu-4dvar: Coupling the data-driven weather forecasting model with 4d variational assimilation." arXiv preprint arXiv:2312.12455 (2023).

---

### Official Review · Reviewer_dYtv · 2025-03-14

**Overall Recommendation:** 3

**Summary:**

This paper proposes Tensor-Var, a framework combining kernel conditional mean embedding (CME) with 4D-Var to linearize nonlinear dynamics and enable convex optimization in a learned feature space. It addresses the high computational costs and theoretical limitations of traditional 4D-Var and deep learning hybrids. Tensor-Var incorporates deep features (DFs) for scalability and provides theoretical guarantees for consistent assimilation. Experiments on chaotic systems and global weather prediction show Tensor-Var achieves higher accuracy and a 10- to 20-fold speedup over conventional and DL hybrid 4D-Var methods, offering an efficient and theoretically sound approach for data assimilation.

## Update after rebuttal
I appreciate the authors' thorough efforts in addressing my feedback. The explanations and revisions have resolved my concerns, and I maintain my current evaluation of weak accept.

**Claims And Evidence:**

Claim 1: Efficiency and Accuracy Improvements:
The paper asserts that Tensor-Var outperforms conventional and ML-hybrid variational DA baselines in both accuracy and computational efficiency. This claim is supported by experimental results on chaotic systems and global NWP applications, showing significant improvements in accuracy and a notable reduction in computational cost. However, the comparison with state-of-the-art ML-based DA methods (e.g., FengWu-4DVar) is not provided, which could strengthen the claim of superiority.

Claim 2: Handling Incomplete Observations:
The authors claim that Tensor-Var improves robustness and accuracy by deriving an inverse observation operator that incorporates historical data to infer system states. While this is a promising approach, the evidence for its effectiveness is limited to the presented experiments. Broader validation on diverse datasets with varying levels of observation incompleteness would strengthen this claim.

Claim 3: Scalability via Deep Features (DFs):
The paper addresses scalability by learning adaptive DFs to map data into a fixed-dimensional feature space. While this is demonstrated on the 64×32 ERA5 dataset in the experiments, the scalability of the approach for even larger-scale problems (e.g., higher-resolution data) is not thoroughly explored, leaving room for further validation.

[1] FengWu-4DVar: Coupling the Data-driven Weather Forecasting Model with 4D Variational Assimilation

**Essential References Not Discussed:**

[1] Verma, Y., Heinonen, M., & Garg, V. (2024). ClimODE: Climate and weather forecasting with physics-informed neural ODEs. arXiv preprint arXiv:2404.10024.

[2] Vaughan, A., Markou, S., Tebbutt, W., Requeima, J., Bruinsma, W. P., Andersson, T. R., ... & Turner, R. E. (2024). Aardvark weather: end-to-end data-driven weather forecasting. arXiv preprint arXiv:2404.00411.

[3] Han, Tao, et al. "Fengwu-ghr: Learning the kilometer-scale medium-range global weather forecasting." arXiv preprint arXiv:2402.00059 (2024).

**Experimental Designs Or Analyses:**

Evaluation Setup:
All metrics are evaluated 20 times with different initial conditions. However, 20 initial fields are insufficient for data assimilation tasks, as it is common to evaluate over an entire year to ensure robustness and reliability.

Baseline Selection:
The choice of baselines could be more diverse. For example, DABench [1] provides a comprehensive set of advanced AI-based and numerical assimilation ensemble methods that could serve as stronger baselines for comparison.

Evaluation Metrics:
The evaluation relies solely on NRMSE. Including additional metrics, such as latitude-weighted RMSE and ACC, would provide a more comprehensive comparison with state-of-the-art DA methods and offer readers better insights into the performance.

Prediction Model Details:
The details of the prediction model used in the assimilation process are not provided. A clear description of the model's architecture, training, and implementation is necessary for reproducibility and thorough evaluation.

[1] DABench: A Benchmark Dataset for Data-Driven Weather Data Assimilation

**Methods And Evaluation Criteria:**

Yes, the methods and evaluation criteria make sense. Tensor-Var uses kernel embedding and deep features to linearize nonlinear dynamics, addressing computational and non-convex challenges in 4D-Var. Experiments on chaotic systems and global NWP applications demonstrate improved accuracy and efficiency compared to conventional and ML-hybrid baselines

**Other Comments Or Suggestions:**

See each part above.

**Other Strengths And Weaknesses:**

Lack of Stability Testing: The authors do not conduct stability tests for the assimilation and forecasting cycle over extended periods. For example, it is unclear whether the proposed DA method can achieve stable predictions in a year-long rolling forecast scenario. This is a critical gap, as long-term stability is essential for operational DA systems.
Lowe
Resolution Limitations: All experiments are conducted at a 32x64 resolution, and it is unclear whether the conclusions scale to higher resolutions. Adding experiments at higher resolutions (e.g., 0.25°) would make the study more solid and practically applicable, as higher resolutions are critical for capturing fine-scale weather phenomena and operational forecasting.

**Questions For Authors:**

Higher-Resolution Experiments:
Can Tensor-Var's performance be validated at higher resolutions (e.g., 0.25°)?
Successful high-resolution results would strengthen practical relevance and increase my score.

Stability Testing for Rolling Forecasts:
Can the authors demonstrate Tensor-Var's stability in year-long rolling forecasts?
Evidence of long-term stability would address a critical limitation and improve my evaluation.

Comparison with State-of-the-Art Methods:
Can Tensor-Var be compared to state-of-the-art methods like FengWu-4DVar?
Superior or comparable performance would validate the method's significance and increase my score.

Addressing these points would significantly enhance the paper's contributions, and I would like to rise my score.

**Relation To Broader Scientific Literature:**

The Tensor-Var framework advances the broader scientific literature by integrating kernel embeddings and deep features to linearize nonlinear dynamics in variational data assimilation (DA), addressing challenges like non-convex optimization and scalability. It builds on prior work in ML-enhanced DA but introduces theoretical guarantees and scalable deep features, achieving significant improvements in efficiency (10- to 20-fold speedup) and accuracy. This bridges gaps in traditional and ML-hybrid DA methods, offering a robust solution for real-time applications like numerical weather prediction.

**Theoretical Claims:**

Yes, the theoretical seems correct.

---

> ### Author Rebuttal · Authors · 2025-04-01
>
> We greatly appreciate valuable and constructive suggestions from Reviewer dYtv. Your efforts have significantly contributed to improving the quality of our paper. Below, we provide detailed responses to each of your comments.
>
> - **Higher-resolution experiment**
>
>   We would like to clarify that the experiments in Section 4.3 are conducted at a resolution of **240×121**, which is roughly 14 times higher than the 64×32 resolution used in Section 4.2.
>
> - **Compare with sota ML-based DA methods & Baseline selections**
>
>    We understand the reviewer’s concern about comparing with state-of-the-art ML-based DA methods, particularly FengWu-4DVar. To address this, we conducted additional evaluations of FengWu-4DVar in two weather data assimilation experiments. We used the official implementation from the [GitHub repository](https://github.com/OpenEarthLab/FengWu-4DVar). Training and evaluation settings follow our implementation as described in Sections 4.2 and 4.3.
>   - We have summarized the updated results for Figure 2 (page 7) for Section 4.2 into the following table for clarity.
>   - For the experiment with higher resolution, we have also added the additional results of FengWu-4DVar into Figure 3 (page 7). Both updated figures (2 and 3) are available in the [anonymous Google Slides](https://docs.google.com/presentation/d/1SWXKhJL_4mAKVQlY4HAxQTl_79C94m-2TwgtkYIvuNw/edit?usp=sharing), providing a clearer demonstration of the results.
>
>       **Table: Assimilation NRMSE (%) across variables and evaluation time (s).**
>       | Method|z500| t850| q700| u850| v850| Evaluation Time |
>       |-|-|-|-|-|-|-|
>       |Latent 3D-Var|3.10±0.28|3.83±0.28|11.4±0.65|6.40±0.38|6.36±0.36|1.84±0.06|
>       |Latent 4D-Var|3.02±1.12|3.89±1.29|13.79±3.17|6.57±1.07|6.31±0.83|4.90±0.26|
>       |FengWu 4D-Var|2.80±0.23|2.88±0.16|5.00±0.43|5.51±0.19|5.40±0.24|1.81±0.11|
>       |Tensor-Var|**1.79±0.11**|**1.94±0.09**|**4.77±0.19**|**3.19±0.14**|**3.47±0.15**|**0.83±0.02**|
>
>       The results demonstrate that **Tensor-Var outperforms FengWu-4DVar in terms of both accuracy and computational efficiency**. Similar results are observed in higher resolution experiments. We provide result figures along with detailed discussions in the shared Google Slides for the reviewer's potential interest.
>
> - **Prediction Model Details**
>
>   In Tensor-Var, predictions are made using the fitted linear operator $\mathcal{C}_{S^+|S}$ without the need for additional learnable models. To improve clarity, we have included a demonstrative figure of the fitting procedure in the revised manuscript's Appendix C and made it available in the Google Slides.
>
> - **Stability Test**
>
>   We thank reviewer for highlighting the importance of long-term stability. In response, we added a long-term rollout stability test with the following key settings:
>
>   - Spatial Resolution: 240×121.
>   - Meteorological Variables: z500, t850, q700, u850, v850.
>   - Time Step: 6 hours per step.
>   - Assimilation Window Length: 5 steps.
>   - Test Horizon: One year (January 1, 2018 - December 31, 2018).
>   - Leading Time: 7 days.
>   - others: Training and evaluation settings follow Section 4.3.
>
>   The result figures can be found in the Google Slides. These results demonstrate that Tensor-Var consistently controls the estimation error over one-year horizon. To benchmark performance, we also include the results of FengWu 4D-Var. Our experiments show that **Tensor-Var achieves better error control with DA compared to FengWu 4D-Var**. However, Tensor-Var rollouts unstablely after 800 steps, where FengWu demonstrates more stable long-term rollouts.
>
>   Tensor-Var’s long-term instability likely stems from linear dynamics, which only capture short-term evolution accurately. In contrast, transformer model in FengWu better handles long-term evolution. We value the reviewer’s insight and will work on improvng Tensor-Var’s stability while maintaining DA efficiency and accuracy.
>
> - **Different metrics**
>
>    We thank reviewer for suggesting using different metrics. To address this, we evaluated FengWu 4D-Var (considered cutting-edge) and Tensor-Var using the anomaly correlation coefficient (ACC) at a resolution of $240*121$. Other settings follow Section 4.3 Figure 3. The results demonstrate that **Tensor-Var outperforms FengWu 4D-Var in ACC** as well. The corresponding figures can be found in the shared Google Slides.
>
>
> - **Clarification of Figure 10**
>
>   We thank reviewer for carefully reviewing our manuscript. The reported metric is *normalized RMSE (NRMSE)*, and we have clarified this in the revised manuscript in Appendix D.3.
>
> - **Suggested References**
>
>   We thank for the suggested references. After careful review, we have properly cited them in the revised manuscript and discussed their relevance.
>
> In the end, we would like to thank reviewer dYtv again for the valuable feedback and invested efforts.  We believe that the revisions address your concerns and improve the overall confidence to our work.

---

### Official Review · Reviewer_Qu2S · 2025-03-17

**Overall Recommendation:** 4

**Summary:**

This paper introduces 4d Tensor-Var, a framework for performing data assimilation with learnable features. Building upon CME to parameterize the state space model, the authors propose two versions of constructing the features space.

**Claims And Evidence:**

I would recommend reorganizing the presentation: currently, the method section is a mix of literature review and modification based on past works. This format creates a barrier to accessing the method. A diagram or an algorithm block (like the ones in Appendix C with more clarification) would be ideal to present the method.

Notations are also confusing: For example, the authors defined the CME (conditional mean embedding) as a mapping from the observation space ($\mathcal{O}$) to latent space ($\mathcal{S}$) on page 3, but later on, when they introduced state feature, the notation they adopted seem to indicate that the CME embedding is from one-step forward feature space $\mathcal{S}^{+}$ to $\mathcal{S}$. This inconsistency without explanation greatly impacts the presentation quality.

Another example we could find is regarding the definition of feature space $z_t$, and the feature mapping function $\phi_{\theta_{\mathcal{S}}}$ which maps from $\mathbb{R}^{n_s}$ to $\mathbb{R}^{d_s}$. The authors did not discuss at all what $n_s$ and $d_s$ mean, and the feature $z_t$ is not introduced and suddenly applied to the loss function in Eqn. 5.

**Essential References Not Discussed:**

I will recommend adding a discussion with the following literature:

KalmanNet: Neural Network Aided Kalman Filtering for Partially Known Dynamics (https://ieeexplore.ieee.org/document/9733186)

Data Assimilation with Machine Learning Surrogate Models: A Case Study with FourCastNet (https://arxiv.org/abs/2405.13180)

**Experimental Designs Or Analyses:**

I do like the empirical studies they perform, as well as the intuitive diagram they provided in Figure 1.

Besides the current results, I am curious about the comparison between Algorithms 1 and 2: how do fixed kernel features compare to deep features?

**Methods And Evaluation Criteria:**

The dataset looks good to me: The authors conducted experiments on chaotic PDE data (KS, Lorenz systems) as well as real weather data (ERA5).

Moreover, I would be curious to see how the training time compares between different methods.

**Other Comments Or Suggestions:**

N/A

**Other Strengths And Weaknesses:**

N/A

**Questions For Authors:**

Please check the "claims" part. I do think improving the clarity of this draft is important.

**Relation To Broader Scientific Literature:**

I think the field of research is important.

**Theoretical Claims:**

Yes, I've checked their theoretical claims. However, because of the ambiguity in the method, I am not sure if I am capable of fully accessing their theoretical claims.

---

> ### Author Rebuttal · Authors · 2025-04-01
>
> We sincerely thank reviewer Qu2S for your thoughtful review and constructive feedback on our manuscript. Your comments have been invaluable in improving the clarity and presentation of our work. Below, we address each of your comments.
>
> - **Reorganize the method section**
>
>     Thanks reviewer’s suggestion, we have improved clarity in the methods section as follows:
>
>     - Added a dedicated Related Work section to clearly distinguish the literature review from our methodological contributions.
>     - Included a diagram illustrating the training procedure for CME opeartor $\mathcal{C}_{S^+|S}$ (can be accessed in the [anonymous Google Slides](https://docs.google.com/presentation/d/1SWXKhJL_4mAKVQlY4HAxQTl_79C94m-2TwgtkYIvuNw/edit?usp=sharing))
>     - Moved the algorithm (previously Algorithm 1) to the main text (end of Section 3.1), revising it to highlight key steps.
>
> - **Clarification of Notations**
>
>    Thank you for carefully reviewing our manuscript. We would like to clarify the following:
>
>    - We do not state the CME operator as a mapping from the observation space $\mathcal{O}$ to latent space $\mathcal{S}$. Instead, we defined it as $\mathcal{C}_{S|O}\colon\mathbb{H}_O\rightarrow\mathbb{H}_S$, where $\mathbb{H}$ with subscript denotes the corresponding feature (latent) spaces.
>
>       The forward CME operator is consistently expressed as $\mathbb{H}_{S^+}\rightarrow\mathbb{H}_{S}$ later in the manuscript.
>
>    - Algorithm 2 (Appendix C) focuses on Tensor-Var with predetermined kernel features, without learnable parameters $\theta$ as in deep features.
>
>    - We realize the notations in our manuscript may be unclear. To improve clarity, we have added a dedicated table in the Appendix of the updated manuscript. You can also find it in the shared Google Slides above.
>
> - **Predetermined kernel features compare to deep features**
>
>     We would like to clarify that we conducted an ablation study to compare different feature dimensions and the Gaussian kernel (Section 4.4, Table 3). Details are in Appendix D.5. Additionally, we tested other kernel features, including Matern-1.5 and Matern-2.5.
>
>   **Table: Comparison of features using NRMSE (%) for state estimation accuracy.**
>   | Features/Experiments  | Lorenz-96 $n_s=40$ | Lorenz-96 $n_s=80$|
>   |-----------------------|----------------------|----------------------|
>   | $d_s=20$            | 16.7±2.1              | 17.3±2.6              |
>   | $d_s=40$            | 14.1±1.3              | 16.8±2.2              |
>   | $d_s=60$            | 8.3±0.9               | 11.4±0.9              |
>   | $d_s=120$          | 9.7±0.7               | **9.0**±0.9              |
>   | Gaussian Kernel       | 8.4±0.5               | 14.3±1.4              |
>   | Matern-1.5            | 9.6±0.8               | 13.2±1.1              |
>   | Matern-2.5            | **7.2±0.2**           | 9.7±**0.7**           |
>
>   In $n_s=40$, Matern-2.5 shows the lowest mean and standard deviation (std). In $n_s=80$, the deep feature with $d_s=120$ outperforms other methods, despite a slightly higher std than Matern-2.5. These results indicate fixed kernels are more reliable at low dimensions, while deep features excel as dimensionality increases. For high-dimensional weather forecasting, fixed kernel features often scale poorly and struggle with structured data. Thus, we propose deep features to extend Tensor-Var for broader applications.
>
> - **Comparing Training Time Across Methods**
>
>   We thank reviewer for the interest in training time. To address this, we measured the training time (per epoch) under identical configurations (NVIDIA RTX 4090) for the experiments in Section 4.2. These results are summarised in the following table and are included in the revised manuscript (Appendix D).
>
>   |Methods|Latent 3D-Var|Latent 4D-Var|FengWu 4D-Var|Tensor-Var|
>   |-|-|-|-|-|
>   |Training time (min)|0.75±0.06|0.88±0.41|11.05±1.65|5.32±1.07 (forward), 4.17±0.31 (inverse)|
>
>     Latent 3D and 4D-Var are computationally efficient as they only consist of convolutional and fully connected layers, requiring only one forward pass in training. FengWu 4D-Var, with its transformer-based encoder-decoder, is more time-consuming due to self-attention operations. Tensor-Var’s forward and inverse models are slower than latent DA due to solving least square problems in feature spaces, while the inverse model is faster as no decoding process.
>
> - **Suggested References**
>
>   We appreciate suggested references. After careful review, we agree that they are relevant to our study. We have properly cited them in the updated manuscript.
>
> We thank reviewer Qu2S again for invested efforts and constructive feedback, which helped improve the manuscript. We have carefully addressed all concerns and made substantial revisions to improve clarity. We look forward to any further feedback.

---

### Decision · Program_Chairs · 2025-05-01

**Decision:**

Accept (poster)

**Comment:**

This paper introduces Tensor-Var, a framework that uses kernel CME to linearize the dynamics in 4D-Var (a data assimilation approach common in meteorology) to yield a more tractable and efficient optimization problem. Reviewers were generally impressed with the methodological contributions, highlighting the potential for impact. While reviewers raised several concerns about the manuscript, almost all of these were addressed during the rebuttal period. These include updated experiments comparing with SOTA ML-based methods, stability tests, comparisons of different kernel functions, and comparisons of training times. I recommend acceptance of this work assuming all of these additions are included in the final revision, and _strongly recommend_ that the comparisons to other SOTA ML methods are included in the main body of the paper.